# Prototype-Based Test-Time Adaptation of Vision-Language Models

**Zhaohong Huang** [1]   **Yuxin Zhang** [1]   **Wenjing Liu** [1]   **Fei Chao** [1]   **Rongrong Ji** [1]

## Abstract

Test-time adaptation (TTA) has emerged as a promising paradigm for vision–language models (VLMs) to bridge the distribution gap between pre-training and test data. Recent works have focused on backpropagation-free TTA methods that rely on cache-based designs, but these introduce two key limitations. First, inference latency increases as the cache grows with the number of classes, leading to inefficiencies in large-scale settings. Second, suboptimal performance occurs when the cache contains insufficient or incorrect samples. In this paper, we present Prototype-Based Test-Time Adaptation (PTA), an efficient and effective TTA paradigm that uses a set of class-specific knowledge prototypes to accumulate knowledge from test samples. Particularly, knowledge prototypes are adaptively weighted based on the zero-shot class confidence of each test sample, incorporating the sample's visual features into the corresponding class-specific prototype. It is worth highlighting that the knowledge from past test samples is integrated and utilized solely in the prototypes, eliminating the overhead of cache population and retrieval that hinders the efficiency of existing TTA methods. This endows PTA with extremely high efficiency while achieving state-of-the-art performance on 15 image recognition benchmarks and 4 robust point cloud analysis benchmarks. For example, PTA improves CLIP's accuracy from 65.64% to 69.38% on 10 cross-domain benchmarks, while retaining 92% of CLIP's inference speed on large-scale ImageNet-1K. In contrast, the cache-based TDA achieves a lower accuracy of 67.97% and operates at only 50% of CLIP's inference speed. Our code is available at https://github.com/hzhxmu/PTA.

[1]Key Laboratory of Multimedia Trusted Perception and Efficient Computing, Ministry of Education of China, Xiamen University, 361005, P.R. China. Correspondence to: Rongrong Ji <rrji@xmu.edu.cn>.

*Proceedings of the 43rd International Conference on Machine Learning*, Seoul, South Korea. PMLR 306, 2026. Copyright 2026 by the author(s).

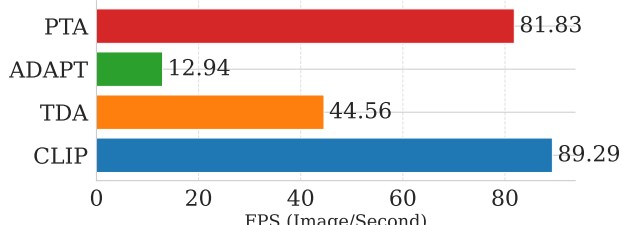

*Figure 1.* Inference speed comparison on the large-scale ImageNet-1K (Deng et al., 2009) shows that our method achieves efficiency comparable to the original CLIP (Radford et al., 2021a), while outperforming cache-based methods such as TDA (Karmanov et al., 2024) and ADAPT (Zhang et al., 2025). All experiments are conducted on a single NVIDIA RTX 3090 GPU.

## 1. Introduction

Vision–language models (VLMs), such as CLIP (Radford et al., 2021a), have demonstrated remarkable zero-shot transferability by aligning visual and textual representations in a shared embedding space. Despite this success, VLMs often suffer from significant performance degradation when deployed in real-world scenarios, due to inevitable distribution shifts between pretraining data and target environments. To mitigate this issue, prior works (Zhou et al., 2022b; Chen et al., 2023; Gao et al., 2024; Zhang et al., 2022) have explored adapting VLMs to downstream tasks using few-shot labeled data. However, these approaches rely heavily on high-quality, task-specific annotations, which are costly to obtain and often unavailable in practical applications.

Test-time adaptation (TTA) has recently emerged as a promising strategy that adapts VLMs using only unlabeled test data, thereby mitigating distribution shifts and reducing reliance on labeled supervision. Existing TTA methods for VLMs can be broadly categorized into two paradigms based on whether test-time backpropagation is required. The first paradigm relies on backpropagation-based adaptation, which updates text prompts (Shu et al., 2022; Feng et al., 2023) or prediction biases (Huang et al., 2025) using random augmentations and carefully designed objectives (*e.g.*, entropy minimization). However, such methods incur substantial computational overhead due to costly backpropagation, limiting their applicability in resource-constrained scenarios (Zhang et al., 2022). For example, test-time prompt tuning (TPT) (Shu et al., 2022) requires 20GB of memory

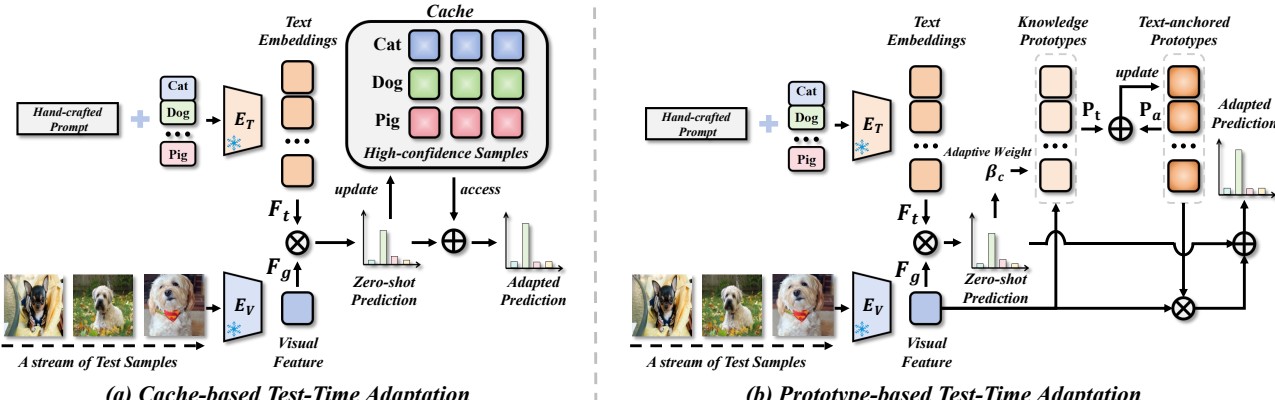

*Figure 2.* Illustration of (a) Cache-Based Test-Time Adaptation and (b) our proposed Prototype-based Test-Time Adaptation (PTA). Unlike cache-based methods that maintain and query a confidence-filtered subset of test samples, PTA introduces a set of class-specific knowledge prototypes to continuously accumulate information from all test samples. By performing adaptation directly at the prototype level, PTA avoids the loss of informative samples and the sparsity issue at early stages, while remaining highly efficient in large-scale settings.

for inference on a single image. Consequently, increasing attention has been given to backpropagation-free TTA methods (Karmanov et al., 2024; Zhang et al., 2024c; 2025; Guan et al., 2025; Zhou et al., 2025), which improve test-time performance by reusing information from previously seen test samples to refine model predictions, all without the need for backpropagation. Most existing backpropagation-free TTA methods adopt cache-based designs. The cache serves as an external memory that enables the model to access previously observed examples during inference. Specifically, test samples are treated as a data stream, and sample features are stored in class-conditioned cache slots with a fixed capacity using zero-shot pseudo-labels (see Figure 2(a)). During the stream, low-entropy instances are retained while high-entropy ones are discarded to update the cache.

Despite the effectiveness of cache-based mechanisms, they still introduce several inherent limitations. First, populating and retrieving the cache introduce non-negligible inference latency, which accumulates as the test stream grows. Although the cache capacity is bounded on a per-class basis, the overall cache size grows with the number of categories, leading to increasing cost in large-scale settings. For example, when scaling to large class spaces such as ImageNet-1K (Deng et al., 2009), the inference speed of cache-based ADAPT (Zhang et al., 2025) drops to only 14.5% of that of the original CLIP (Radford et al., 2021a), severely limiting its practicality in latency-sensitive scenarios (see Figure 1). Second, model adaptation is tightly coupled with the cache state. When the cache is empty or sparsely populated at the beginning of the test stream, the model has little usable information, resulting in suboptimal performance. For example, on Oxford Pets (Parkhi et al., 2012), cache-based methods only surpass the baseline after observing more than 1,000 test samples (see Figure 3). As more samples arrive,

the cache may become effective, but this improvement is not guaranteed to be stable. Under limited capacity and heuristic filtering rules, informative samples can be discarded, while incorrect ones may persist and accumulate errors. As a result, performance can even degrade over time, as observed on Caltech101 (Fei-Fei et al., 2004) (see Figure 3).

In response to the above drawbacks, this paper presents Prototype-based Test-Time Adaptation (PTA), a cache-free and backpropagation-free approach that maintains a set of class-specific representative vectors, termed knowledge prototypes. As shown in Figure 2(b), PTA utilizes these knowledge prototypes to accumulate knowledge from test samples in real-time, playing a unique role in improving TTA performance. Particularly, PTA assigns each test sample an adaptive weight based on its zero-shot class confidence, and accumulate its visual features into the corresponding class-specific knowledge prototype. This enables PTA to leverage all test samples, circumventing the loss of correct sample information and ensuring immediate knowledge accumulation without waiting for a sufficient number of samples. It is worth highlighting that PTA achieves extremely low latency, as the accumulation and utilization of test sample knowledge occur solely within the prototypes, eliminating the need for cache updates and retrievals as required by previous methods. Consequently, as shown in Figures 1 and 3, PTA achieves a well-balanced trade-off between performance and efficiency of TTA.

We evaluate PTA on 15 image recognition datasets across cross-domain and out-of-distribution (OOD) benchmarks, where it achieves notable performance gains over state-of-the-art methods. PTA can also be easily extended to other visual modalities beyond images, where it outperforms both the baseline and the cache-based Point-Cache (Sun et al.,

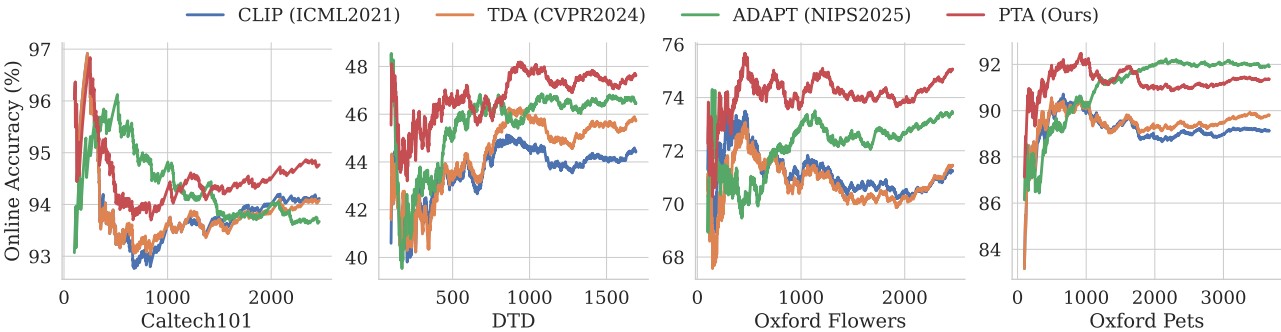

*Figure 3.* Online accuracy of different methods on 4 cross-domain benchmarks (Fei-Fei et al., 2004; Maji et al., 2013; Nilsback & Zisserman, 2008; Parkhi et al., 2012). The x-axis represents the cumulative number of test samples encountered in the data stream. To account for the initial cold-start period, online accuracy is computed after the first 100 samples. Notably, all methods adopt the same hand-crafted prompts from CLIP (Radford et al., 2021a) for fair comparison.

2025) on 4 challenging robust point cloud analysis benchmarks. In terms of efficiency, PTA reaches 81.83 FPS on ImageNet-1K (Deng et al., 2009), significantly outperforming the cache-based TDA and ADAPT, which achieve 44.56 FPS and 12.94 FPS, respectively. Our contributions in this paper include:

- We propose Prototype-based Test-Time Adaptation (PTA), which accumulates knowledge from test samples via a set of class-specific knowledge prototypes.

- PTA innovatively performs backpropagation-free TTA solely in the prototypes, drastically reducing the TTA latency of existing methods.

- We provide comprehensive evaluations on 15 image recognition benchmarks and 4 robust point cloud analysis benchmarks, demonstrating that PTA achieves an excellent balance between performance and efficiency.

## 2. Related Works

### 2.1. Contrastive Vision-Language Models

Contrastive vision-language models (Radford et al., 2021a; Jia et al., 2021; Xue et al., 2023; 2024) achieve effective zero-shot visual understanding by learning rich cross-modal correlations from large-scale visual-text pairs. However, the inherent distribution gap between pretraining data and downstream test environments may cause substantial performance degradation. Existing methods (Zhou et al., 2022a; Khattak et al., 2023; Zhang et al., 2022; Gao et al., 2024) typically address this issue by collecting a small amount of labeled downstream data to fine-tune VLMs for task adaptation. For instance, CoOp (Zhou et al., 2022b) replaces handcrafted prompts with learnable context tokens for task adaptation, while TextRefiner (Xie et al., 2025) further incorporates internal image knowledge into optimized vectors. While effective, these methods require labeled data and incur extra

training costs. In contrast, in this work, we investigate test-time adaptation for VLMs, which enables models to adapt to downstream data without requiring any labeled data.

### 2.2. Test-Time Adaptation for VLMs

Test-time adaptation (TTA) aims to improve the performance of pretrained models on test data without relying on additional labeled supervision. Recently, this paradigm has been explored in the context of vision–language models (VLMs) (Zhang et al., 2024a; Zanella & Ben Ayed, 2024; Huang et al., 2025; Zhou et al., 2025). Pioneering works such as TPT (Shu et al., 2022) adapt learnable text prompts by enforcing consistency across multiple augmented views of each test sample, while DiffTPT (Feng et al., 2023) improves generalization by incorporating augmented data generated by a diffusion model. Although effective, these methods rely on computationally expensive backpropagation at test time. This requirement significantly limits their applicability in latency-sensitive or resource-constrained scenarios. To address this issue, recent efforts have shifted towards backpropagation-free TTA methods, which typically rely on cache-based designs (Zhang et al., 2024c; Karmanov et al., 2024; Zhang et al., 2025; Guan et al., 2025). For example, TDA (Karmanov et al., 2024) maintains positive and negative caches to store high-confidence and high-uncertainty samples, respectively, and refines predictions by retrieving information from the cache. ADAPT (Zhang et al., 2025) maintains a knowledge bank to cache high-confidence samples and performs test-time adaptation by explicitly modeling class-conditional Gaussian distributions. However, existing cache-based TTA methods commonly rely on maintaining and querying an external memory whose size scales with the number of classes or stored samples, inevitably introducing increased inference latency in large-scale scenarios. In addition, their adaptation performance strongly depends on the reliability of cached samples, as incorrect or insufficient entries can lead to suboptimal performance.

In this work, we propose PTA that introduces a set of class-specific knowledge prototypes to accumulate knowledge from test samples. Crucially, PTA accumulates and utilizes the knowledge from previously observed samples solely through the prototypes, thereby circumventing the inherent limitations of cache-based designs in terms of both efficiency and performance.

## 3. Method

### 3.1. Background

**Contrastive Vision Language Models.** By jointly embedding images and text into a shared latent space, contrastive vision language models (Radford et al., 2021a; Jia et al., 2021; Alayrac et al., 2022) achieve rich cross-modal representations. Without loss of generality, we adopt CLIP (Radford et al., 2021b) as the foundational model for our methodological demonstration. CLIP is composed of a visual encoder $\mathbf{E_V}$ and a text encoder $\mathbf{E_T}$, which extract features from images and text, respectively. During training, a contrastive loss (Chen et al., 2020) is used to maximize similarity between matched image-text pairs while minimizing it for non-matching pairs.

At test time, given an input image $\boldsymbol{x}$, the visual encoder produces a visual feature $\boldsymbol{F}_g = \mathbf{E_V}(\boldsymbol{x}) \in \mathbb{R}^d$, where $d$ is the feature dimension. For a specific downstream task with $C$ classes, each class is incorporated into a hand-crafted prompt, yielding the corresponding text embeddings $\{\boldsymbol{F}_t^c\}_{c=1}^C$, where $\boldsymbol{F}_t^c \in \mathbb{R}^d$ denotes the text embedding of the class-specific text input. The prediction probability of image $\boldsymbol{x}$ belonging to class $y_c$ is then computed from the cosine similarity between the visual feature and the text embeddings, expressed as:

$$p_{\text{CLIP}}(y_c|\boldsymbol{x}) = \frac{\exp\left(\cos\left(\boldsymbol{F}_g, \boldsymbol{F}_t^c\right)/\tau\right)}{\sum_{c=1}^C \exp\left(\cos\left(\boldsymbol{F}_g, \boldsymbol{F}_t^c\right)/\tau\right)}, \quad (1)$$

where $cos(\cdot)$ calculates the cosine similarity between vectors, and $\tau$ controls the sharpness of the softmax distribution.

**Cache-based Test-time Adaptation.** As illustrated in Figure 2(a), these methods maintain a fixed-size cache $\boldsymbol{Q}_{\text{cache}} \in \mathbb{R}^{MC \times d}$, with $M$ entries per class and $C$ classes, to store features extracted from test samples. At test time, a new sample $\boldsymbol{x}$ with pseudo-label $c$ is inserted into the cache only if the cache is not yet full, or if it satisfies a predefined filtering criterion, e.g., having lower prediction entropy than the highest-entropy entry in the class-specific cache $\boldsymbol{Q}_{\text{cache}}^c \in \mathbb{R}^{M \times d}$. In this way, the cache acts as an external memory that allows the model to retrieve and leverage previously observed test samples during inference. Although cache-based methods achieve backpropagation-free test-time adaptation, their efficiency critically depends on the cache capacity $M$ and the number of classes $C$. As a

result, inference latency inevitably increases in large-scale settings. Moreover, cache-based adaptation requires a warm-up phase before becoming effective, during which sparse cached samples lead to suboptimal performance, while strict filtering heuristics may further discard informative samples and allow incorrect ones to persist and bias the adaptation process.

### 3.2. Prototype-based Test-Time Adaptation

In this work, we propose Prototype-based Test-Time Adaptation (PTA) as a way to address the above cache-based method limitations. The core contribution of PTA lies in introducing a set of class-specific knowledge prototypes, which accumulate knowledge from streaming test samples in real-time. As illustrated in Figure 2(b), knowledge prototypes adaptively integrate the visual features of each test sample according to their class confidence, thereby accumulating knowledge of the test domain over time. We describe the detailed implementation of PTA below.

We begin by zero-initializing a set of class-specific knowledge prototypes $\mathbf{P}_t \in \mathbb{R}^{C \times d}$, where each prototype $\mathbf{P}_t^c \in \mathbb{R}^d$ corresponds to the knowledge prototype for class $c$ and shares the same feature dimension as the CLIP text prototypes $\mathbf{F}_t$:

$$\mathbf{P}_t = \{\mathbf{P}_t^1, \mathbf{P}_t^2, \dots, \mathbf{P}_t^C\}. \quad (2)$$

At test time, given an incoming sample $\boldsymbol{x}$, we obtain its visual feature $\boldsymbol{F}_g$ and the prediction confidence $p_{\text{CLIP}}(y \mid \boldsymbol{x})$ over all $C$ classes according to Eq. 1. The prediction confidence $p_{\text{CLIP}}(y \mid \boldsymbol{x})$ reflects the relevance of a test sample to each class and thus serves as a strong prior for test-time adaptation (Zhang et al., 2022). We treat this confidence as the class-wise contribution score of sample $\boldsymbol{x}$, denoted as $\boldsymbol{s} = p_{\text{CLIP}}(y \mid \boldsymbol{x})$. The objective of PTA is to maintain a set of class-specific knowledge prototypes that progressively accumulate visual features from incoming test samples. Inspired by previous works (Ma et al., 2023), we adopt the Exponential Moving Average (EMA) mechanism, a widely used strategy for stabilizing feature aggregation in streaming settings. In EMA, the decay rate controls how much each incoming sample contributes to the prototype update. Since different test samples exhibit varying class-specific confidence scores, we replace the fixed decay rate with a dynamic one that adapts to each sample's confidence. A straightforward choice is to directly use the raw confidence scores $\boldsymbol{s}$ as the decay rate for prototype updates. However, this can lead to imbalanced knowledge integration, as early high-confidence samples tend to dominate the updates and drive rapid convergence toward a narrow set of features. To mitigate this issue, we apply a bounded and monotonic transformation to the class confidence $s_j$, yielding a smooth adaptive weight $\beta_j$ for each class $j$:

$$\beta_j = 1 - e^{-\boldsymbol{s}_j/h}, \quad (3)$$

*Table 1.* Comparison of PTA in Cross-Datasets Generalization using ViT-B/16 (Dosovitskiy, 2020) as the backbone. **Bold** values indicate the best performance on backpropagation-free TTA methods. For a fair comparison, we unify the handcrafted prompts used by all backpropagation-free TTA methods by adopting the prompt templates described in CLIP (Radford et al., 2021a), where † denotes results reproduced using the officially released code. Other results are taken from the original papers, using consistent prompt templates.

| Method | Venue | Flowers | DTD | Pets | Cars | UCF101 | Caltech | Food101 | SUN397 | Aircraft | EuroSAT | **Average** |
|---|---|---|---|---|---|---|---|---|---|---|---|---|
| CLIP-ViT-B/16† | ICML2021 | 71.34 | 44.39 | 89.10 | 65.91 | 66.72 | 94.12 | 86.08 | 66.31 | 24.78 | 47.69 | 65.64 |
| *Backpropagation-Based Test-Time Adaptation* | | | | | | | | | | | | |
| TPT | NIPS2022 | 68.98 | 47.75 | 87.79 | 68.50 | 68.04 | 94.16 | 84.67 | 65.50 | 24.78 | 42.44 | 65.10 |
| DiffTPT | ICCV2023 | 70.10 | 47.00 | 88.22 | 67.01 | 62.67 | 92.49 | 87.23 | 65.74 | 25.60 | 43.13 | 65.47 |
| DPE | NIPS2024 | 75.07 | 54.20 | 91.14 | 67.31 | 70.44 | 94.81 | 86.17 | 70.07 | 28.95 | 55.79 | 69.40 |
| HisTPT | NIPS2024 | 71.20 | 48.90 | 89.10 | 69.20 | 70.10 | 94.50 | 89.30 | 67.20 | 26.90 | 49.70 | 67.60 |
| GS-Bias | ICML2025 | 71.94 | 46.10 | 90.38 | 67.33 | 67.59 | 94.60 | 86.09 | 67.40 | 26.49 | 52.42 | 67.03 |
| *Backpropagation-Free Test-Time Adaptation* | | | | | | | | | | | | |
| BoostAdapter | NIPS2024 | 71.66 | 45.69 | 89.51 | **69.30** | 71.93 | 94.77 | **87.17** | 68.09 | **27.45** | 61.22 | 68.68 |
| MTA† | CVPR2024 | 71.34 | 44.68 | 89.37 | 66.43 | 67.33 | 94.32 | 86.25 | 66.82 | 25.23 | 47.75 | 65.95 |
| TDA† | CVPR2024 | 71.50 | 45.09 | 89.37 | 67.03 | 70.82 | 94.08 | 86.27 | 67.81 | 25.32 | 62.51 | 67.97 |
| BCA† | CVPR2025 | 70.77 | 44.62 | 89.02 | 66.80 | 67.12 | 93.91 | 86.04 | 66.57 | 24.87 | 49.23 | 65.90 |
| TCA | ICCV2025 | 73.33 | 46.16 | 89.53 | 65.33 | 72.38 | 93.69 | 85.31 | 65.92 | 24.87 | 70.43 | 68.69 |
| SCA† | NIPS2025 | 74.45 | **47.72** | 89.91 | 67.57 | 71.34 | 93.91 | 86.16 | 68.84 | 25.03 | 52.57 | 67.75 |
| ADAPT† | NIPS2025 | 73.28 | 46.57 | **91.58** | 68.56 | 70.58 | 93.39 | 85.72 | 69.09 | 25.59 | **63.06** | 68.74 |
| PTA | Ours | **75.23** | 47.70 | 91.06 | 68.55 | **73.12** | **94.81** | 86.44 | **69.21** | 26.10 | 61.57 | **69.38** |

where $h$ is a temperature parameter that controls the update smoothness. Subsequently, we use the adaptive decay rate $\beta_j$ to inject each test sample's visual feature $\boldsymbol{F}_g$ into the corresponding class-specific knowledge prototype $\mathbf{P}_t$ via a dynamic EMA:

$$\mathbf{P}_t^j = (1 - \beta_j) \, \mathbf{P}_t^j + \beta_j \boldsymbol{F}_g. \tag{4}$$

As successive test samples are incorporated, the knowledge prototypes $\mathbf{P}_t$ continuously accumulate knowledge from the incoming data. However, relying solely on $\mathbf{P}_t$, which contains only visual knowledge, for prediction may cause the prototypes to drift away from CLIP's pretrained text-image joint embedding space, in which text embeddings serve as essential anchors for robust zero-shot generalization (Shu et al., 2022; Feng et al., 2023). To maintain this critical alignment, we introduce a set of text-anchored prototypes $\mathbf{P}_a \in \mathbb{R}^{C \times d}$, initialized from the original CLIP text embeddings $\boldsymbol{F}_t$. A simple interpolation between $\mathbf{P}_t$ and $\mathbf{P}_a$ is then performed to ensure that the adapted prototypes remain grounded in the pretrained semantic space:

$$\mathbf{P}_a = (1 - w) \, \mathbf{P}_t + w \mathbf{P}_a, \tag{5}$$

where $w$ is the coefficient hyperparameter. Analogous to Eq. 1, we first compute the prototype-based logit $p_{\texttt{Prototype}}(y|\boldsymbol{x})$ using as the text-anchored prototypes:

$$p_{\texttt{Prototype}}(y_c|\boldsymbol{x}) = \frac{\exp\left(\cos\left(\boldsymbol{F}_g, \mathbf{P}_a^c\right)/\tau\right)}{\sum_{c=1}^{C} \exp\left(\cos\left(\boldsymbol{F}_g, \mathbf{P}_a^c\right)/\tau\right)}. \tag{6}$$

Finally, the final PTA prediction is obtained by combining the prototype-based logits with the original zero-shot

CLIP logits, which preserves CLIP's foundational zero-shot capability at the logit level:

$$p_{\texttt{PTA}}(y|\boldsymbol{x}) = p_{\texttt{CLIP}}(y|\boldsymbol{x}) + p_{\texttt{Prototype}}(y|\boldsymbol{x}). \tag{7}$$

It is important to highlight that PTA can be seamlessly generalized to diverse tasks such as robust point cloud analysis (Uy et al., 2019; Ren et al., 2022) by simply substituting image features with point cloud embeddings.

## 4. Experiments

### 4.1. Experimental Setup

**Datasets.** To comprehensively evaluate the effectiveness of our method, we conduct experiments on 15 image recognition datasets and 4 robust point cloud analysis datasets.

For image recognition, we follow prior work (Shu et al., 2022; Karmanov et al., 2024; Zhang et al., 2024a) and adopt two widely adopted TTA benchmarks: cross-domain generalization and out-of-distribution (OOD) generalization. Under the cross-domain generalization benchmark, we evaluate performance on 10 diverse datasets spanning a wide spectrum of classification tasks. These datasets include Flowers102 (Nilsback & Zisserman, 2008), OxfordPets (Parkhi et al., 2012), StanfordCars (Krause et al., 2013), FGVC-Aircraft (Maji et al., 2013), Food101 (Bossard et al., 2014), EuroSAT (Helber et al., 2019), UCF101 (Soomro, 2012), DTD (Cimpoi et al., 2014), SUN397 (Sun et al., 2020), and Caltech101 (Fei-Fei et al., 2004). For the OOD generalization benchmark, we further test our approach on 4 challenging OOD variants of ImageNet-1K (Deng et al., 2009): ImageNetV2 (Recht et al., 2019), ImageNet-Sketch (Wang

*Table 2.* Comparison of PTA in OOD Generalization using ViT-B/16 (Dosovitskiy, 2020) as the backbone. **Bold** values indicate the best performance on backpropagation-free TTA methods. For a fair comparison, we unify the handcrafted prompts used by all backpropagation-free TTA methods by adopting the prompt templates described in CLIP (Radford et al., 2021a), where † denotes results reproduced using the officially released code. Average and OOD Average denote the mean accuracy over all 5 datasets and the 4 OOD datasets (excluding ImageNet), respectively.

| Method | ImageNet | ImageNet-A | ImageNet-V2 | ImageNet-R | ImageNet-S | Average | OOD Average |
|---|---|---|---|---|---|---|---|
| CLIP-ViT-B/16 | 66.73 | 47.87 | 60.86 | 73.98 | 46.09 | 57.20 | 59.11 |
| *Backpropagation-Based Test-Time Adaptation* | | | | | | | |
| TPT | 68.98 | 54.77 | 63.45 | 77.06 | 47.94 | 62.44 | 60.81 |
| DiffTPT | 70.30 | 55.68 | 65.10 | 75.00 | 46.80 | 62.28 | 60.52 |
| GS-Bias | 70.57 | 56.61 | 64.62 | 80.49 | 50.33 | 64.52 | 63.01 |
| DPE | 71.91 | 59.63 | 65.44 | 80.40 | 52.26 | 65.93 | 64.43 |
| *Backpropagation-Free Test-Time Adaptation* | | | | | | | |
| MTA | 70.08 | 58.06 | 64.24 | 78.33 | 49.61 | 64.06 | 62.56 |
| TDA | 69.51 | 60.11 | 64.67 | 80.24 | 50.54 | 65.01 | 63.89 |
| BCA | 70.22 | 61.14 | 64.90 | 80.72 | 50.87 | 65.37 | 64.16 |
| BoostAdapter† | 69.41 | **62.90** | **65.03** | 80.72 | 51.31 | **65.87** | **64.99** |
| SCA† | 70.21 | 61.13 | 64.01 | **80.93** | 51.11 | 65.48 | 64.30 |
| ADAPT† | **70.55** | 60.87 | 64.44 | 80.85 | **51.49** | 65.64 | 64.41 |
| PTA | 70.28 | 61.15 | 64.85 | 80.79 | 51.00 | 65.61 | 64.45 |

et al., 2019), ImageNet-A (Hendrycks et al., 2021b), and ImageNet-R (Hendrycks et al., 2021a).

For robust point cloud analysis, we follow the Point-Cache setup (Sun et al., 2025) and adopt 4 corrupted point cloud benchmarks: ModelNet-C (Ren et al., 2022) and 3 corrupted variants of ScanObjectNN (Uy et al., 2019). ModelNet-C introduces 7 atomic corruption types, including adding global and local outliers, removing global structures and local parts, as well as rotation, scaling, and jittering. Following the same protocol, these atomic corruptions are applied to the 3 variants of ScanObjectNN to construct their corrupted counterparts.

**Baselines.** We compare our proposed method, PTA, with several state-of-the-art approaches across 15 image recognition benchmarks. The comparison includes zero-shot CLIP (Radford et al., 2021a) and 13 TTA methods, including 5 backpropagation-based TTA (TPT (Shu et al., 2022), DiffTPT (Feng et al., 2023), DPE (Zhang et al., 2024a), HisTPT (Zhang et al., 2024b), and GS-Bias (Huang et al., 2025)) and 7 backpropagation-free TTA (Boost-Adapter (Zhang et al., 2024c), MTA (Zanella & Ben Ayed, 2024), TDA (Karmanov et al., 2024), BCA (Zhou et al., 2025), TCA (Wang et al., 2025), SCA (Guan et al., 2025) and ADAPT (Zhang et al., 2025)), all tailored for vision-language models. Additionally, we compare our method with ULIP (Xue et al., 2023), a representative zero-shot method for point cloud recognition, and Point-Cache (Sun et al., 2025), a cache-based TTA method, on 4 robust point cloud analysis benchmarks.

**Implementation details.** In line with prior work (Shu et al., 2022; Karmanov et al., 2024), we use publicly available

pretrained CLIP models (Radford et al., 2021a) for image recognition, with ViT-B/16 (Dosovitskiy, 2020) backbones as the visual encoder and Transformer (Vaswani, 2017) as the text encoder. Additionally, we employ the pretrained ULIP model (Xue et al., 2023) for robust point cloud analysis tasks. The batch size is set to 1 to meet the requirements of streaming processing. It is important to highlight that PTA adopts the ensemble prompting strategy described in CLIP (Radford et al., 2021a), which is not generated by a large language model (LLM), ensuring a fairer evaluation of TTA methods. For our PTA, we set $h = 20$ (Eq. 3) and $w = 0.01$ (Eq. 5) across all benchmarks. We report top-1 accuracy as the evaluation metric, with results averaged over 3 runs using different random seeds. All experiments are conducted on an NVIDIA 3090 GPU.

### 4.2. Comparisons with State-of-the-art

**Results on the Cross-Domain Generalization.** Table 1 reports the results on cross-domain benchmarks. Without performing any backpropagation at test time, PTA achieves substantial improvements over backpropagation-based prompt optimization methods, outperforming TPT (Shu et al., 2022) and DiffTPT (Feng et al., 2023) by 4.28% and 3.91%, respectively. Notably, PTA achieves performance comparable to DPE (Zhang et al., 2024a), a cache-based prototype learning method that relies on gradient updates and incorporates LLM-generated prompts as handcrafted priors. This comparison highlights that PTA serves as a more efficient and principled prototype-based TTA approach, achieving competitive performance without gradient updates, external caches, and additional LLM priors. Compared with other backpropagation-free TTA methods, PTA consistently

*Table 3.* Results under non-iid streaming scenarios with different $\gamma$, where smaller $\gamma$ indicates stronger non-iidness. In the *Separate* setting, classes arrive sequentially.

| Scenario | Method | ImageNet | Flowers | DTD | Pets | Cars | UCF101 | Caltech | Food101 | SUN397 | Aircraft | EuroSAT | **Average** |
|---|---|---|---|---|---|---|---|---|---|---|---|---|---|
| - | CLIP | 66.6 | 70.7 | 43.5 | 89.1 | 65.6 | 67.5 | 93.2 | 85.9 | 62.5 | 24.7 | 48.3 | 65.1 |
| *Low* | TDA | 68.3 | 72.5 | 45.5 | 89.6 | 66.9 | 71.0 | 93.4 | 86.1 | 66.0 | 25.4 | 60.6 | 67.7 |
| $(\gamma = 0.1)$ | PTA | 67.2 | 74.2 | 46.3 | 90.5 | 67.2 | 71.1 | 93.7 | 86.3 | 66.9 | 25.6 | 57.1 | **67.8** |
| *Medium* | TDA | 68.2 | 72.6 | 45.2 | 89.3 | 66.5 | 70.1 | 93.5 | 85.8 | 66.2 | 25.2 | 56.5 | 67.1 |
| $(\gamma = 0.01)$ | PTA | 67.5 | 73.3 | 45.8 | 90.6 | 67.7 | 70.9 | 93.8 | 86.1 | 65.8 | 25.5 | 54.8 | **67.4** |
| *High* | TDA | 67.9 | 72.5 | 45.1 | 89.0 | 66.3 | 69.7 | 93.6 | 85.5 | 65.1 | 25.1 | 55.3 | 66.8 |
| $(\gamma = 0.001)$ | PTA | 68.1 | 73.3 | 45.6 | 90.6 | 68.2 | 70.9 | 93.8 | 85.8 | 66.6 | 25.5 | 54.1 | **67.5** |
| *Separate* | TDA | 67.4 | 72.3 | 45.0 | 88.9 | 65.9 | 69.6 | 93.6 | 85.2 | 64.6 | 24.9 | 55.3 | 66.6 |
| | PTA | 68.5 | 73.2 | 45.6 | 90.6 | 68.5 | 71.1 | 93.7 | 85.6 | 66.7 | 25.5 | 53.9 | **67.5** |

achieves the strongest performance. For example, PTA improves the average accuracy by 0.70%, 1.41%, 1.63%, and 0.64% over cache-based methods including Boost-Adapter (Zhang et al., 2024c), TDA (Karmanov et al., 2024), SCA (Guan et al., 2025), and ADAPT (Zhang et al., 2025), respectively. These results demonstrate that our prototype-based TTA exhibits superior robustness under domain shifts, as it can more effectively exploit a broader set of test samples rather than relying on a filtered cache subset.

**Results on the OOD Generalization.** We further evaluate PTA on out-of-distribution (OOD) generalization benchmarks. As reported in Table 2, PTA achieves strong average OOD performance among existing TTA methods. In particular, PTA outperforms the backpropagation-based GS-Bias (Huang et al., 2025) by 1.44%, suggesting that streaming knowledge accumulation is more effective for OOD generalization than adapting to each test sample independently. PTA also consistently improves upon the backpropagation-free TDA (Karmanov et al., 2024) across datasets, demonstrating its robustness to complex distribution shifts in OOD environments. Although BoostAdapter (Zhang et al., 2024c) achieves higher OOD accuracy by expanding the cache with augmented views, its gains come with additional test-time computation. In contrast, PTA adopts a cache-free paradigm, achieving efficient TTA through compact prototype-level knowledge accumulation.

**Results on Non-IID Test Streams.** To further evaluate the robustness of PTA under highly imbalanced streaming scenarios, we construct non-iid test streams on the 10 cross-domain datasets and ImageNet-1K using a Dirichlet distribution (Zanella et al., 2025). Specifically, the inter-batch class correlation is controlled by the Dirichlet parameter $\gamma$, where a smaller $\gamma$ indicates stronger non-iidness. We also consider an extreme *Separate* setting, where classes arrive sequentially. To reduce evaluation variance, we generate 100 tasks for each configuration and report the averaged accuracy with a batch size of 128. For this more challenging setting, we

adopt a more conservative configuration with $h = 1000$ and $w = 0.001$. As shown in Table 3, PTA remains stable as the test stream becomes increasingly non-iid. Across all non-iid settings, PTA consistently outperforms both CLIP and TDA in terms of average accuracy. Notably, the advantage of PTA becomes more pronounced under stronger distribution shifts, especially in the *High* and *Separate* settings. These results indicate that PTA is robust to non-iid distribution shifts and degrades more gracefully than the cache-based TDA under challenging streaming conditions.

**Results on Robust Point Cloud Analysis.** Beyond image classification, we extend PTA to the robust point cloud analysis task to demonstrate its broad applicability. As shown in Table 4, we evaluate its robustness across 4 corrupted point cloud benchmarks (Ren et al., 2022; Uy et al., 2019), where PTA significantly improves the zero-shot performance of ULIP (Xue et al., 2023) across all datasets and corruption types. For instance, at corruption severity level 2, PTA boosts performance by +7.61% on ModelNet-C, +3.71% on ScanObjectNN (OBJ-ONLY), +5.70% on ScanObjectNN (OBJ-BG), and +5.43% on ScanObjectNN (hardest). Additionally, we further validate PTA's robustness across 5 corruption severity levels in Figure 4. It is noteworthy that our prototype-based PTA consistently outperforms the cache-based Point-Cache (Sun et al., 2025) across all levels of corruption severity. For example, on the ModelNet-C, PTA achieves performance improvements of +1.8%, +1.3%, +1.4%, +1.3%, and +1.6% across corruption levels 0 to 4, respectively. These results further highlight that, even when extended to point cloud modalities, prototype-based TTA maintains superior robustness.

**Efficiency Analysis.** We also report an efficiency comparison on the large-scale ImageNet-1K (Deng et al., 2009). As shown in Table 5, backpropagation-based methods are computationally expensive. For instance, the FPS of TPT (Shu et al., 2022) is only 1% of the original CLIP speed, with memory usage increasing $27\times$. In contrast,

*Table 4.* Comparison of recognition accuracy on ModelNet-C (Ren et al., 2022) and 3 corrupted variants of ScanObjectNN-C (Uy et al., 2019) that includes 7 types of corruptions. Results are reported for a corruption severity level of 2. Each clean point cloud contains 1024 points. The last column is the average across the 7 types of corruptions.

| Method | Corruption Type | | | | | | | Average |
| | Add Global | Add Local | Drop Global | Drop Local | Rotate | Scale | Jitter | |
|---|---|---|---|---|---|---|---|---|
| *ModelNet-C* | | | | | | | | |
| ULIP | 33.55 | 43.92 | 54.70 | 50.89 | 55.27 | 50.20 | 44.08 | 47.52 |
| Point-Cache | 46.15 | 47.85 | 59.16 | 56.00 | **61.47** | 55.35 | 48.91 | 53.70 |
| PTA | **48.78** | **51.17** | **61.54** | **56.69** | 61.42 | **56.44** | **49.88** | **55.13** |
| *ScanObjectNN (OBJ-ONLY)* | | | | | | | | |
| ULIP | 31.50 | 34.77 | 51.29 | 38.38 | 48.36 | 44.58 | 36.83 | 40.82 |
| Point-Cache | 32.01 | 38.04 | **54.56** | 45.27 | 50.95 | 45.96 | 39.24 | 43.72 |
| PTA | **34.76** | **39.76** | 53.53 | **45.78** | **51.81** | **46.47** | **39.59** | **44.53** |
| *ScanObjectNN (OBJ-BG)* | | | | | | | | |
| ULIP | 27.19 | 25.82 | 45.61 | 34.25 | 40.96 | 40.10 | 30.98 | 34.99 |
| Point-Cache | 28.23 | 30.12 | 48.71 | 40.45 | 43.55 | 40.28 | **34.42** | 37.97 |
| PTA | **32.36** | **30.98** | **52.15** | **42.00** | **48.53** | **46.64** | 32.19 | **40.69** |
| *ScanObjectNN (hardest)* | | | | | | | | |
| ULIP | 19.26 | 18.39 | 30.99 | 23.91 | 27.48 | 26.34 | 21.44 | 23.97 |
| Point-Cache | 23.46 | **22.69** | 34.70 | 31.75 | 33.00 | 28.28 | **25.05** | 28.42 |
| PTA | **24.18** | 22.07 | **38.41** | **32.51** | **34.21** | **30.78** | 23.63 | **29.40** |

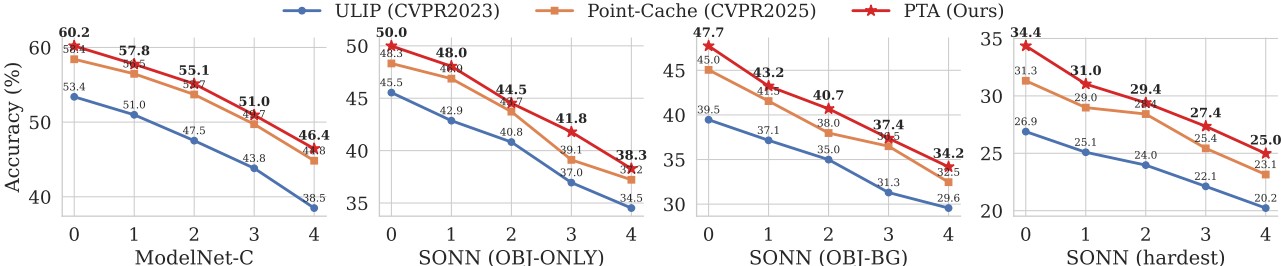

*Figure 4.* Comparison of recognition accuracy on ModelNet-C (Ren et al., 2022) and 3 corrupted variants of ScanObjectNN (SONN) (Uy et al., 2019), across 7 corruption types at 5 severity levels. Each clean point cloud contains 1024 points. Results are averaged over the 7 corruption types.

backpropagation-free methods generally offer higher efficiency. However, while cache-based backpropagation-free methods reduce memory usage, they do not achieve significant reductions in inference latency due to the unavoidable overhead of populating and retrieving external caches in large-scale settings. For example, TDA (Karmanov et al., 2024) operates at only 50% of the original CLIP speed. In comparison, our PTA achieves exceptional efficiency, reaching 92% of the original CLIP speed.

### 4.3. Ablation Study

In this section, we comprehensively analyze PTA from two perspectives: the sensitivity to key hyperparameters and the robustness against early high-confidence erroneous updates. Unless otherwise specified, all ablation studies are conducted on 11 benchmark datasets, including ImageNet and the 10 datasets in the cross-domain benchmark.

**The effects of $h$.** We study the effect of the temperature parameter $h$ in updating the knowledge prototypes. As shown in Figure 5, performance initially improves with

increasing $h$, but then starts to decline beyond a certain point. Smaller values of $h$ lead to overly aggressive prototype updates, which can result in biased knowledge accumulation. In contrast, larger values of $h$ cause underfitting, leading to suboptimal performance. This pattern is consistent across all datasets, indicating that $h$ does not require fine-tuning for specific datasets, thereby demonstrating the robustness of PTA.

**The effects of $w$.** Table 6 analyzes the impact of $w$ in Eq. 5. Setting $w = 0$ causes catastrophic failure, confirming that purely visual prototypes without textual anchoring collapse into non-discriminative representations. In contrast, PTA is remarkably robust across $w \in [0.0001, 0.01]$. However, excessively large $w$ (*e.g.*, 0.1) suppresses the model's ability to integrate domain-specific knowledge, leading to performance degradation. These results demonstrate that minimal textual regularization is sufficient to stabilize adaptation.

**Robustness to early high-confidence erroneous updates.** We examine whether PTA is vulnerable to early high-confidence erroneous updates. Specifically, we move con-

*Table 5.* A comparison of efficiency on ImageNet-1K (Deng et al., 2009) is shown, with all results reproduced from the officially released code. All experiments were conducted on a single NVIDIA RTX 3090 GPU.

| Method | FPS (Image/Second) | Memory |
|---|---|---|
| CLIP | 89.3 (100%) | 732 Mib |
| *Backpropagation-Based TTA* | | |
| TPT | 1.1 (1%) | 19997 Mib |
| DPE | 3.8 (4%) | 6538 Mib |
| GS-Bias | 10.3 (12%) | 1308 Mib |
| *Backpropagation-Free TTA* | | |
| MTA | 10.6 (12%) | 1448 Mib |
| TDA | 44.6 (50%) | 762 Mib |
| BoostAdapter | 37.5 (42%) | 834 Mib |
| SCA | 57.9 (65%) | 794 Mib |
| ADAPT | 12.9 (14%) | 892 Mib |
| PTA | **81.8** (92%) | **744 Mib** |

*Table 6.* Ablation study on the hyperparameter $w$ in Eq. 5.

| $w$ ($h = 20$) | 0 | 0.0001 | 0.001 | 0.01 | 0.1 |
|---|---|---|---|---|---|
| ImageNet | 0.10 | 70.12 | 70.14 | 70.28 | 68.00 |
| Cross-Domain | 2.56 | 69.35 | 69.35 | 69.38 | 65.60 |
| Average | 1.33 | 69.74 | 69.75 | 69.83 | 66.80 |

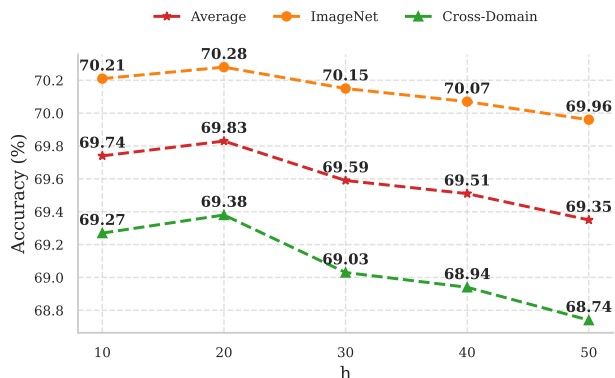

*Figure 5.* Ablation study on the hyperparameter $h$ in Eq. 3.

*Table 7.* Robustness to early high-confidence erroneous updates. We place confidently misclassified samples at the beginning of the stream and vary the poisoned prefix size.

| Dataset | 1 | 5 | 10 | All | Normal |
|---|---|---|---|---|---|
| ImageNet | 70.3 | 70.2 | 70.3 | 70.3 | 70.3 |
| Cross-Domain | 69.3 | 69.4 | 69.4 | 69.3 | 69.4 |
| Average | 69.8 | 69.8 | 69.9 | 69.8 | 69.9 |

fidently misclassified samples, whose confidence is larger than 0.8, to the beginning of the test stream. We vary the poisoned prefix size among 1, 5, 10, and all available samples, and compare the results with the normal stream order. As shown in Table 7, PTA maintains stable performance across all settings, with the average accuracy ranging from 69.8 to 69.9. This shows that PTA is insensitive to early high-confidence errors and avoids noticeable error amplification. The reason is that PTA relies on soft class-wise confidence weighting for prototype updates, instead of using the top-1 pseudo-label as a hard update target.

### 4.4. Supplementary Experiments

In the appendix, we provide additional experimental analyses to further substantiate the effectiveness of PTA. These include evaluations on different backbones (A.1), orthogonality analysis with existing TTA methods (A.2), robustness to varying data stream orders (A.3), efficiency analysis on additional datasets (A.4), the impact of adaptive decay rate $\beta$ in Eq. 3 (A.5), robustness to different prompt templates (A.6), performance on ImageNet-C (A.7), as well as further robust point cloud analysis results (A.8).

## 5. Limitations and Future Work

We further discuss unexplored limitations of our proposed PTA method. First, test-time adaptation entails both benefits and risks, since the model can change its behavior after deployment. Although PTA achieves efficient adaptation through prototype-level knowledge accumulation, the prototypes may still absorb spurious correlations from local test streams, potentially causing unpredictable behavior or bias amplification in sensitive scenarios. Meanwhile, the EMA-style prototype accumulation may retain outdated information under highly non-stationary streams, which can slow adaptation to abrupt distribution shifts. Therefore, exploring fairness-aware and drift-aware prototype updates, such as adaptive forgetting or prototype reset, remains an important direction for future work.

## 6. Conclusion

In this work, we propose Prototype-Based Test-Time Adaptation (PTA), a novel backpropagation-free TTA paradigm designed to overcome the efficiency and performance bottlenecks of existing cache-based methods. By accumulating knowledge directly within class-specific prototypes via an adaptive weighting mechanism, PTA eliminates the overhead of external memory maintenance while ensuring stable and immediate adaptation. Extensive evaluations across image recognition and robust point cloud analysis benchmarks demonstrate that PTA achieves state-of-the-art performance with exceptional inference speed, reaching 92% of original CLIP's efficiency on ImageNet-1K. These results establish PTA as a practical and robust solution for deploying vision-language models in latency-sensitive scenarios.

## Acknowledgement

This work is supported by the National Key Research and Development Program of China (No. 2025YFE0113500), the National Science Fund for Distinguished Young Scholars (No. 62525605), and the National Natural Science Foundation of China (No. U25B2066).

## Impact Statement

This paper proposes PTA, a backpropagation-free Test-Time Adaptation (TTA) method for vision-language models that strikes a balance between performance and efficiency. The TTA community may benefit from the insights and findings presented in our research. There are many potential societal consequences of our work, none which we feel must be specifically highlighted here.

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

# A. Appendix

*Table 8.* Comparison of PTA in Cross-Datasets Generalization using ResNet50 (He et al., 2016) as the backbone. **Bold** values indicate the best performance on backpropagation-free TTA methods. For a fair comparison, we unify the handcrafted prompts used by all backpropagation-free TTA methods by adopting the prompt templates described in CLIP (Radford et al., 2021a), where † denotes results reproduced using the officially released code. Other results are reported directly from the original papers.

| Method | Venue | Flowers | DTD | Pets | Cars | UCF101 | Caltech | Food101 | SUN397 | Aircraft | EuroSAT | **Average** |
|---|---|---|---|---|---|---|---|---|---|---|---|---|
| CLIP-ResNet50† | ICML2021 | 66.10 | 42.20 | 85.83 | 56.55 | 61.41 | 87.91 | 76.34 | 61.11 | 17.22 | 37.42 | 59.21 |
| *Backpropagation-Based Test-time Adaptation* | | | | | | | | | | | | |
| TPT | NIPS2022 | 62.69 | 40.84 | 84.49 | 58.46 | 60.82 | 87.02 | 74.88 | 61.46 | 17.58 | 28.33 | 57.66 |
| DiffTPT | ICCV2023 | 63.53 | 40.72 | 83.40 | 60.71 | 62.67 | 86.89 | 79.21 | 62.72 | 17.60 | 41.04 | 59.85 |
| DPE | NIPS2024 | 67.60 | 50.18 | 85.97 | 59.26 | 61.98 | 90.83 | 77.83 | 64.23 | 19.80 | 41.67 | 61.93 |
| *Backpropagation-Free Test-time Adaptation* | | | | | | | | | | | | |
| TDA† | CVPR2024 | **68.70** | 43.62 | 86.15 | 57.39 | 63.81 | **89.70** | **77.74** | 62.37 | **17.40** | 41.23 | 60.81 |
| BCA† | CVPR2025 | 66.26 | 42.38 | 85.45 | 58.13 | 61.80 | 87.75 | 77.39 | 61.36 | 17.28 | 29.04 | 58.68 |
| SCA† | NIPS2025 | 67.72 | **46.69** | 86.43 | 59.23 | 66.03 | 88.40 | 76.72 | 63.17 | 17.01 | 48.51 | 61.99 |
| ADAPT† | NIPS2025 | 67.97 | 45.57 | **87.76** | **59.68** | 63.49 | 88.24 | 76.35 | **63.77** | 17.22 | 47.44 | 61.75 |
| PTA | Ours | 68.49 | 44.98 | 86.67 | 59.35 | **66.43** | 88.64 | 76.83 | 62.86 | **17.44** | **51.63** | **62.33** |

## A.1. Evaluation on Different Backbones

In this section, we provide a quantitative evaluation of PTA using ResNet50 (He et al., 2016) as the backbone to further validate its architectural robustness on the cross-domain generalization benchmark. As shown in Table 8, PTA consistently achieves state-of-the-art results among backpropagation-free methods, reaching a top average accuracy of 62.33%. Notably, while competing methods like TDA (Karmanov et al., 2024) and ADAPT (Zhang et al., 2025) excel on specific datasets, PTA demonstrates superior overall stability and generalization, particularly on challenging benchmarks such as EuroSAT (Helber et al., 2019) and UCF101 (Soomro, 2012). These results underscore that the effectiveness of PTA is not reliant on a specific large-scale architecture like ViT (Dosovitskiy, 2020), but rather stems from its prototype-based knowledge accumulation mechanism, which remains robust and efficient across different backbone configurations.

*Table 9.* Performance enhancement of existing TTA methods when integrated with our PTA. We evaluate the orthogonality of PTA by incorporating it into various TTA frameworks, including backpropagation-based (GS-Bias (Huang et al., 2025)), backpropagation-free (MTA (Zanella & Ben Ayed, 2024)), and cache-based (TDA (Karmanov et al., 2024)) methods.

| Method | ImageNet | Flowers | DTD | Pets | Cars | UCF101 | Caltech | Food101 | SUN397 | Aircraft | EuroSAT | **Average** |
|---|---|---|---|---|---|---|---|---|---|---|---|---|
| GS-Bias | 70.57 | 71.94 | 46.10 | 90.38 | 67.33 | 67.59 | 94.60 | 86.09 | 67.40 | 26.49 | 52.42 | 67.35 |
| **+PTA** | **71.00** | **74.50** | **48.46** | **91.36** | **69.12** | **72.01** | **95.05** | **86.83** | **69.55** | **26.49** | **61.31** | **69.61** |
| MTA | 70.08 | 71.34 | 44.68 | 89.37 | 66.43 | 67.33 | 94.32 | 86.25 | 66.82 | 25.23 | 47.75 | 66.33 |
| **+PTA** | **71.27** | **73.33** | **50.12** | **91.03** | **69.33** | **71.98** | **94.97** | **86.82** | **69.48** | **26.88** | **57.44** | **69.33** |
| TDA | 69.51 | 71.50 | 45.09 | 89.37 | 67.03 | 70.82 | 94.08 | 86.27 | 67.81 | 25.32 | **62.51** | 68.07 |
| **+PTA** | **70.31** | **75.07** | **47.34** | **90.81** | **68.98** | **73.99** | **94.73** | **86.57** | **69.32** | **26.40** | 61.88 | **69.58** |

## A.2. Orthogonality to Existing TTA Frameworks

To further demonstrate the versatility of our approach, we investigate whether PTA can serve as a complementary module to existing TTA paradigms. As illustrated in Table 9, we integrate PTA into 3 representative categories: (i) GS-Bias (Huang et al., 2025), which relies on backpropagation, (ii) MTA (Zanella & Ben Ayed, 2024), a backpropagation-free baseline, and (iii) TDA (Karmanov et al., 2024), a typical cache-based method. The results reveal a consistent and significant performance boost across all configurations when PTA is incorporated. Notably, PTA improves the average accuracy of GS-Bias (Huang et al., 2025) and MTA (Zanella & Ben Ayed, 2024) by 2.26% and 3.00%, respectively, and further refines the performance of the cache-based TDA (Karmanov et al., 2024). This consistent improvement underscores the strong orthogonality of our prototype-based knowledge accumulation. It suggests that PTA provides unique benefits by maintaining stable, long-term class-specific prototype representations. These findings establish PTA not only as a standalone TTA solution but also as a universal enhancement module capable of bolstering diverse TTA methods.

*Table 10.* Performance of PTA across different random data sequences. We report the results of 3 independent experiments (Exp. 1-3) conducted with different random seeds, where the test samples in the data stream are shuffled into different temporal sequences. The low standard deviation (Std.) across all benchmarks demonstrates that PTA is highly robust to the order of incoming test samples.

| Method | ImageNet | Flowers | DTD | Pets | Cars | UCF101 | Caltech | Food101 | SUN397 | Aircraft | EuroSAT |
|---|---|---|---|---|---|---|---|---|---|---|---|
| Exp.1 | 70.26 | 75.12 | 47.83 | 91.06 | 68.48 | 73.13 | 94.77 | 86.46 | 69.25 | 26.12 | 61.62 |
| Exp.2 | 70.26 | 75.32 | 47.72 | 91.10 | 68.57 | 73.00 | 94.81 | 86.46 | 69.25 | 25.97 | 61.43 |
| Exp.3 | 70.30 | 75.25 | 47.55 | 91.02 | 68.60 | 73.23 | 94.85 | 86.40 | 69.13 | 26.21 | 61.66 |
| **Avg.** | 70.28 | 75.23 | 47.70 | 91.06 | 68.55 | 73.12 | 94.81 | 86.44 | 69.21 | 26.10 | 61.57 |
| **Std.** | ±0.02 | ±0.10 | ±0.14 | ±0.04 | ±0.06 | ±0.11 | ±0.04 | ±0.03 | ±0.06 | ±0.12 | ±0.12 |

## A.3. Robustness to Data Ordering

In real-world streaming scenarios, the sequence in which test samples arrive is often unpredictable. To evaluate the stability of our method under such conditions, we conduct 3 independent runs of PTA using different random seeds to shuffle the input data stream. As presented in Table 10, PTA exhibits remarkable consistency across all datasets regardless of the sample order. Quantitatively, the standard deviation (Std.) remains exceptionally low across all benchmarks, with the average fluctuation being less than ±0.15%. This stability is particularly noteworthy given the streaming nature of the task, where early-stage updates typically exert a foundational influence on the adaptation trajectory. The results confirm that our adaptive weighting and prototype accumulation mechanisms effectively mitigate the risk of being biased by specific sample sequences or early-stage outliers. Consequently, PTA provides a reliable and predictable performance gain, establishing it as a robust solution for diverse and non-stationary real-world data streams.

*Table 11.* Statistics of datasets and comparison of test-time adaptation latency.

| Dataset | ImageNet | Flowers | DTD | Pets | Cars | UCF101 | Caltech | Food101 | SUN397 | Aircraft | EuroSAT |
|---|---|---|---|---|---|---|---|---|---|---|---|
| **#Classes** | 1000 | 102 | 47 | 37 | 196 | 101 | 100 | 101 | 397 | 100 | 10 |
| **#Samples** | 50,000 | 2,463 | 1,692 | 3,669 | 8,041 | 3,783 | 2,465 | 30,300 | 19,850 | 3,333 | 8,100 |
| **TDA Time** | 1122s | 30s | 22s | 41s | 109s | 43s | 32s | 360s | 323s | 41s | 91s |
| **PTA Time** | **611s** | **29s** | **19s** | **41s** | **94s** | **41s** | **28s** | **335s** | **268s** | **40s** | **90s** |

## A.4. Computational Efficiency Analysis

A critical bottleneck for cache-based TTA methods is the accumulation of inference latency as the number of classes and test samples increases. As detailed in Table 11, we compare the adaptation latency of PTA against the cache-based TDA (Karmanov et al., 2024) across 11 benchmarks. While both methods show comparable efficiency on small-scale datasets (*e.g.*, EuroSAT (Helber et al., 2019)), the latency gap widens significantly as the category space expands. For example, on ImageNet-1K (Deng et al., 2009), which involves 1,000 classes and 50,000 samples, TDA (Karmanov et al., 2024) requires 1,122 seconds to complete the stream due to the increasing overhead of cache maintenance and similarity-based retrieval. In contrast, PTA reduces the total time to 611 seconds, achieving a nearly $2\times$ speedup. This efficiency stems from PTA's cache-free design, where knowledge is directly accumulated into prototypes through real-time updates, regardless of the class count. These results confirm that PTA is uniquely suited for real-world, large-scale streaming applications where low latency is as crucial as classification accuracy.

*Table 12.* Comparative analysis of knowledge accumulation weights. We evaluate the performance of PTA using raw confidence scores ($s$) versus our proposed adaptive weights ($\beta$) for prototype updates.

| Weight | ImageNet | Flowers | DTD | Pets | Cars | UCF101 | Caltech | Food101 | SUN397 | Aircraft | EuroSAT |
|---|---|---|---|---|---|---|---|---|---|---|---|
| $s$ | 0.10 | 0.49 | 2.13 | 2.73 | 0.55 | 1.16 | 5.27 | 0.99 | 0.25 | 0.99 | 11.11 |
| $\beta$ | 70.28 | 75.23 | 47.70 | 91.06 | 68.55 | 73.12 | 94.81 | 86.44 | 69.21 | 26.10 | 61.57 |

*Table 13.* Comprehensive comparison across different prompt templates. We evaluate PTA and competitive methods under 3 levels of prompt complexity: (i) Vanilla (a single fixed template), (ii) CLIP Template (average of multiple hand-crafted templates), and (iii) GPT (fine-grained class descriptions). † denotes results reproduced using the officially released code. Other results are reported directly from the original papers.

| Method | Venue | Flowers | DTD | Pets | Cars | UCF101 | Caltech | Food101 | SUN397 | Aircraft | EuroSAT | **Average** |
|---|---|---|---|---|---|---|---|---|---|---|---|---|
| *Vanilla (*`a photo of a [class]`*) (Radford et al., 2021a)* | | | | | | | | | | | | |
| CLIP† | ICML2021 | 67.15 | 44.39 | 88.01 | 65.29 | 65.00 | 92.94 | 85.41 | 62.59 | 23.79 | 41.44 | 63.60 |
| TPT | NIPS2022 | 68.98 | 47.75 | 87.79 | 68.50 | 68.04 | 94.16 | 84.67 | 65.50 | 24.78 | 42.44 | 65.10 |
| DiffTPT | ICCV2023 | 70.10 | 47.00 | 88.22 | 67.01 | 62.67 | 92.49 | 87.23 | 65.74 | 25.60 | 43.13 | 65.47 |
| TDA† | CVPR2024 | 69.87 | 46.16 | 88.74 | 66.68 | 68.62 | 93.43 | 85.93 | 65.99 | 24.36 | 53.02 | 66.28 |
| SCA† | NIPS2025 | 72.72 | 48.58 | 89.04 | 67.17 | 70.00 | 93.31 | 85.72 | 67.32 | 24.54 | 53.48 | 67.19 |
| PTA | Ours | 72.63 | 48.64 | 89.42 | 68.46 | 70.63 | 93.83 | 86.11 | 67.22 | 24.84 | 55.46 | **67.72** |
| *CLIP Template (Radford et al., 2021a)* | | | | | | | | | | | | |
| CLIP† | ICML2021 | 71.34 | 44.39 | 89.10 | 65.91 | 66.72 | 94.12 | 86.08 | 66.31 | 24.78 | 47.69 | 65.64 |
| BoostAdapter | NIPS2024 | 71.66 | 45.69 | 89.51 | 69.30 | 71.93 | 94.77 | 87.17 | 68.09 | 27.45 | 61.22 | 68.68 |
| TDA† | CVPR2024 | 71.50 | 45.09 | 89.37 | 67.03 | 70.82 | 94.08 | 86.27 | 67.81 | 25.32 | 62.51 | 67.97 |
| TDA | CVPR2024 | 71.42 | 47.40 | 88.63 | 67.28 | 70.66 | 94.24 | 86.14 | 67.62 | 23.91 | 58.00 | 67.53 |
| SCA† | NIPS2025 | 74.45 | 47.72 | 89.91 | 67.57 | 71.34 | 93.91 | 86.16 | 68.84 | 25.03 | 52.57 | 67.75 |
| PTA | Ours | 75.23 | 47.70 | 91.06 | 68.55 | 73.12 | 94.81 | 86.44 | 69.21 | 26.10 | 61.57 | **69.38** |
| *GPT (Zhang et al., 2024a)* | | | | | | | | | | | | |
| CLIP† | ICML2021 | 72.76 | 53.25 | 90.19 | 65.95 | 66.69 | 94.24 | 85.85 | 67.93 | 27.21 | 53.60 | 67.77 |
| TDA† | CVPR2024 | 74.02 | 54.61 | 90.95 | 66.80 | 69.79 | 94.48 | 86.12 | 68.92 | 28.59 | 54.88 | 68.92 |
| DPE | NIPS2024 | 75.07 | 54.20 | 91.14 | 67.31 | 70.44 | 94.81 | 86.17 | 70.07 | 28.95 | 55.79 | 69.40 |
| BCA | CVPR2025 | 73.12 | 53.49 | 90.43 | 66.86 | 67.59 | 94.69 | 85.97 | 68.41 | 28.59 | 56.63 | 68.59 |
| SCA† | NIPS2025 | 76.70 | 56.62 | 91.66 | 68.04 | 71.19 | 94.44 | 86.04 | 70.21 | 28.41 | 56.85 | 70.02 |
| SCA | NIPS2025 | 76.09 | 57.09 | 91.44 | 68.49 | 73.43 | 94.85 | 86.09 | 70.27 | 28.50 | 57.16 | 70.34 |
| PTA | Ours | 77.22 | 56.15 | 91.47 | 68.24 | 71.40 | 95.33 | 86.32 | 70.51 | 28.50 | 59.14 | **70.43** |

## A.5. Impact of Adaptive Decay Rate $\beta$

To verify the necessity of the exponential saturation function in Eq. 3, we compare the performance of PTA using raw confidence scores $s$ against our proposed adaptive decay rate $\beta$. As shown in Table 12, relying solely on raw confidence scores $s$ as update weights leads to a catastrophic performance drop, particularly on large-scale datasets like ImageNet (Deng et al., 2009). This experimental evidence confirms our hypothesis that in a streaming environment, raw confidence scores can be excessively polarized. Without the smoothing effect of the saturation function, a few peak-confidence samples can disproportionately dominate the prototype's representation, effectively suppressing the contribution of subsequent informative data. In contrast, by mapping the confidence into a more stable range through $\beta$, PTA ensures a more balanced and equitable knowledge accumulation process.

## A.6. Robustness to Different Prompt Templates

To further investigate the adaptability of PTA, we evaluate its performance under varying degrees of prompt engineering, ranging from a single Vanilla template to descriptive GPT-generated priors. As illustrated in Table 13, PTA consistently achieves the strongest performance across all 3 settings, reaching average accuracies of 67.72%, 69.38%, and 70.43%, respectively. A key observation is that while sophisticated prompts (*e.g.*, GPT-based (Zhang et al., 2024a)) generally enhance the performance of all methods, the performance of several methods is highly sensitive to the prompt quality. For instance, SCA (Guan et al., 2025) exhibits a significant performance gap between Vanilla and GPT settings, whereas PTA remains highly effective even with minimal textual guidance (*e.g.*, Vanilla (Radford et al., 2021a)). This indicates that PTA does not merely rely on rich external linguistic knowledge to bridge the distribution gap. Instead, its strength lies in the principled prototype accumulation, which robustly aligns visual features with the pretrained semantic space regardless of the prompt's granularity.

## A.7. Performance on ImageNet-C

We further evaluate PTA on ImageNet-C (Hendrycks & Dietterich, 2019) to examine its robustness against common corruption shifts. ImageNet-C covers 15 corruption types, including noise, blur, weather, and digital corruptions, which

*Table 14.* Performance comparison on ImageNet-C (Hendrycks & Dietterich, 2019).

| Method | Gaussian | Shot | Impulse | Defocus | Glass | Motion | Zoom | Snow | Frost | Fog | Brightness | Contrast | Elastic | Pixelate | JPEG | **Avg** |
|--------|----------|------|---------|---------|-------|--------|------|------|-------|-----|------------|----------|---------|----------|------|---------|
| CLIP | 13.0 | 14.4 | 13.7 | 24.1 | 15.8 | 25.0 | 26.6 | 32.6 | 30.7 | 37.0 | 55.0 | 17.6 | 13.6 | 33.0 | 33.5 | 25.7 |
| TDA | 9.6 | 11.5 | 11.2 | 24.8 | 14.6 | 23.9 | 25.0 | 35.1 | 32.8 | 38.7 | 56.5 | 16.3 | 14.2 | 38.8 | 33.8 | 25.8 |
| PTA | 12.9 | 15.2 | 13.9 | 24.5 | 15.8 | 25.6 | 26.3 | 33.6 | 32.1 | 37.4 | 56.6 | 17.3 | 14.4 | 33.5 | 35.2 | **26.3** |

introduce diverse distribution shifts beyond the standard ImageNet validation set.

Table 14 shows that PTA achieves the highest average accuracy among the compared methods. Compared with CLIP and TDA, PTA improves the average accuracy by 0.6% and 0.5%, respectively. Although the gains vary across corruption types, PTA performs favorably on a wide range of corruptions, including Shot, Impulse, Motion, Brightness, Elastic, and JPEG. These results suggest that the proposed prototype-level adaptation is effective not only on standard OOD benchmarks, but also under common corruption shifts.

### A.8. Additional robust point cloud analysis results

Tables 15-18 report detailed recognition accuracy under 7 corruption types at severity levels (0, 1, 3, and 4), covering ModelNet-C (Ren et al., 2022) and 3 variants of ScanObjectNN (Uy et al., 2019). Overall, PTA consistently outperforms ULIP (Xue et al., 2023) and Point-Cache (Sun et al., 2025) across datasets and severity levels. Under mild corruptions (severity 0), PTA shows clear advantages on point dropping and geometric transformations, indicating better preservation of discriminative structure. As the severity increases, although all methods experience performance degradation, PTA degrades more gracefully and maintains higher average accuracy, especially on the more challenging ScanObjectNN settings. These results demonstrate that PTA provides stable robustness across varying corruption intensities and types.

*Table 15.* Comparison of recognition accuracy on ModelNet-C (Ren et al., 2022) and 3 corrupted variants of ScanObjectNN-C (Uy et al., 2019) that includes 7 types of corruptions. Results are reported for a corruption severity level of 0. Each clean point cloud contains 1024 points. The last column is the average across the 7 types of corruptions.

| Method | Corruption Type | | | | | | | Avg. |
|--------|------------|-----------|------------|-----------|--------|-------|--------|------|
| | Add Global | Add Local | Drop Global | Drop Local | Rotate | Scale | Jitter | |
| *ModelNet-C* | | | | | | | | |
| ULIP | 45.71 | 51.13 | 55.88 | 56.85 | 56.48 | 53.00 | 54.66 | 53.39 |
| Point-Cache | 52.76 | 55.02 | 60.21 | 61.51 | **64.06** | **58.67** | 56.73 | 58.42 |
| PTA | **56.36** | **56.97** | **63.49** | **64.14** | 63.45 | 57.98 | **59.00** | **60.20** |
| *ScanObjectNN (OBJ-ONLY)* | | | | | | | | |
| ULIP | 35.97 | 39.41 | 49.23 | 47.50 | 50.95 | 48.02 | 47.68 | 45.54 |
| Point-Cache | 40.45 | **42.51** | 52.84 | 51.12 | 53.53 | 47.50 | 50.26 | 48.32 |
| PTA | **42.86** | 40.79 | **54.22** | **53.70** | **54.56** | **51.12** | **52.67** | **49.99** |
| *ScanObjectNN (OBJ-BG)* | | | | | | | | |
| ULIP | 29.43 | 33.22 | 44.58 | 42.51 | 43.72 | 40.28 | 42.51 | 39.46 |
| Point-Cache | 35.28 | 37.35 | 52.84 | 48.71 | 50.09 | 45.09 | 45.96 | 45.05 |
| PTA | **37.52** | **40.28** | **53.53** | **51.81** | **51.46** | **49.23** | **50.26** | **47.73** |
| *ScanObjectNN (hardest)* | | | | | | | | |
| ULIP | 19.88 | 21.86 | 30.92 | 28.56 | 30.40 | 28.49 | 28.14 | 26.89 |
| Point-Cache | 25.95 | 25.95 | 33.66 | 34.63 | 34.14 | 29.94 | **34.98** | 31.32 |
| PTA | **28.52** | **29.42** | **38.65** | **37.37** | **37.61** | **34.14** | 34.77 | **34.35** |

*Table 16.* Comparison of recognition accuracy on ModelNet-C (Ren et al., 2022) and 3 corrupted variants of ScanObjectNN-C (Uy et al., 2019) that includes 7 types of corruptions. Results are reported for a corruption severity level of 1. Each clean point cloud contains 1024 points. The last column is the average across the 7 types of corruptions.

| Method | Corruption Type | | | | | | | Avg. |
|---|---|---|---|---|---|---|---|---|
| | Add Global | Add Local | Drop Global | Drop Local | Rotate | Scale | Jitter | |
| *ModelNet-C* | | | | | | | | |
| ULIP | 38.74 | 47.49 | 55.47 | 54.98 | 56.08 | 52.51 | 51.58 | 50.98 |
| Point-Cache | 49.96 | 50.24 | 60.70 | 58.43 | **62.48** | **58.14** | **55.35** | 56.47 |
| PTA | **51.78** | **53.85** | **62.48** | **61.91** | 62.44 | 57.54 | 54.34 | **57.76** |
| *ScanObjectNN (OBJ-ONLY)* | | | | | | | | |
| ULIP | 33.91 | 37.01 | 51.81 | 42.69 | 49.40 | 44.92 | 40.28 | 42.86 |
| Point-Cache | 38.04 | 40.28 | 54.04 | **50.09** | 52.15 | 44.58 | **48.88** | 46.87 |
| PTA | **39.76** | **40.62** | **56.11** | 49.23 | **53.36** | **49.74** | 47.50 | **48.05** |
| *ScanObjectNN (OBJ-BG)* | | | | | | | | |
| ULIP | 26.33 | 29.09 | 47.16 | 38.55 | 44.23 | 39.93 | 34.77 | 37.15 |
| Point-Cache | 30.46 | 32.53 | 51.12 | 42.69 | 48.02 | 45.27 | 40.79 | 41.55 |
| PTA | **34.08** | **33.73** | **52.67** | **44.06** | **51.12** | **45.78** | **41.14** | **43.23** |
| *ScanObjectNN (hardest)* | | | | | | | | |
| ULIP | 19.01 | 20.54 | 30.71 | 24.98 | 28.80 | 27.31 | 24.29 | 25.09 |
| Point-Cache | 23.28 | 23.28 | 35.32 | 31.96 | 32.41 | 29.60 | **27.00** | 28.98 |
| PTA | **25.57** | **25.33** | **38.27** | **34.00** | **36.75** | **31.05** | 26.27 | **31.03** |

*Table 17.* Comparison of recognition accuracy on ModelNet-C (Ren et al., 2022) and 3 corrupted variants of ScanObjectNN-C (Uy et al., 2019) that includes 7 types of corruptions. Results are reported for a corruption severity level of 3. Each clean point cloud contains 1024 points. The last column is the average across the 7 types of corruptions.

| Method | Corruption Type | | | | | | | Avg. |
|---|---|---|---|---|---|---|---|---|
| | Add Global | Add Local | Drop Global | Drop Local | Rotate | Scale | Jitter | |
| *ModelNet-C* | | | | | | | | |
| ULIP | 29.86 | 41.98 | 52.55 | 47.73 | 51.34 | 49.51 | 33.79 | 43.82 |
| Point-Cache | **45.14** | 47.81 | 57.82 | 51.74 | 55.06 | 53.85 | 36.63 | 49.72 |
| PTA | 44.04 | **49.47** | **58.87** | **53.65** | **57.78** | **54.54** | **38.33** | **50.95** |
| *ScanObjectNN (OBJ-ONLY)* | | | | | | | | |
| ULIP | 28.74 | 32.53 | 47.33 | 34.77 | 39.41 | **44.92** | 30.98 | 36.95 |
| Point-Cache | 29.26 | **33.56** | 49.40 | 36.66 | 44.23 | 42.86 | 37.87 | 39.12 |
| PTA | **33.39** | 33.39 | **53.01** | **43.20** | **47.16** | 44.41 | **37.90** | **41.78** |
| *ScanObjectNN (OBJ-BG)* | | | | | | | | |
| ULIP | 25.13 | 25.13 | 43.72 | 29.09 | 34.08 | 37.52 | 24.44 | 31.30 |
| Point-Cache | 27.54 | 28.06 | 46.30 | 38.04 | 43.89 | 39.93 | **31.67** | 36.49 |
| PTA | **28.57** | **29.43** | **48.36** | **38.97** | **45.61** | **42.00** | 28.74 | **37.38** |
| *ScanObjectNN (hardest)* | | | | | | | | |
| ULIP | 18.15 | 17.28 | 30.29 | 20.58 | 23.66 | 25.85 | 19.01 | 22.12 |
| Point-Cache | 22.03 | 20.54 | 32.51 | 26.16 | 26.79 | 28.24 | 21.72 | 25.43 |
| PTA | **23.42** | **20.61** | **36.02** | **28.70** | **31.12** | **29.35** | **22.21** | **27.35** |

*Table 18.* Comparison of recognition accuracy on ModelNet-C (Ren et al., 2022) and 3 corrupted variants of ScanObjectNN-C (Uy et al., 2019) that includes 7 types of corruptions. Results are reported for a corruption severity level of 4. Each clean point cloud contains 1024 points. The last column is the average across the 7 types of corruptions.

| Method | Corruption Type | | | | | | | Avg. |
|---|---|---|---|---|---|---|---|---|
| | Add Global | Add Local | Drop Global | Drop Local | Rotate | Scale | Jitter | |
| *ModelNet-C* | | | | | | | | |
| ULIP | 26.62 | 38.78 | 45.42 | 41.13 | 44.98 | 48.58 | 23.95 | 38.49 |
| Point-Cache | **42.30** | 42.63 | 50.65 | 44.49 | 49.64 | 54.13 | 27.43 | 44.83 |
| PTA | 41.77 | **46.39** | **54.01** | **47.12** | **49.72** | **55.27** | **30.79** | **46.44** |
| *ScanObjectNN (OBJ-ONLY)* | | | | | | | | |
| ULIP | 28.74 | 30.81 | 44.23 | 30.98 | 37.01 | 43.37 | 26.51 | 34.52 |
| Point-Cache | 31.15 | **32.87** | 46.82 | 36.14 | 35.28 | 45.78 | **32.53** | 37.22 |
| PTA | **31.84** | 31.84 | **47.85** | **36.83** | **40.96** | **46.13** | **32.53** | **38.28** |
| *ScanObjectNN (OBJ-BG)* | | | | | | | | |
| ULIP | 24.61 | 25.65 | 40.10 | 26.68 | 32.53 | 36.32 | 21.00 | 29.56 |
| Point-Cache | 25.82 | **27.37** | **44.92** | 29.09 | 32.53 | 39.76 | **27.71** | 32.46 |
| PTA | **28.57** | **27.37** | **44.92** | **33.56** | **38.04** | **41.14** | 25.65 | **34.18** |
| *ScanObjectNN (hardest)* | | | | | | | | |
| ULIP | 17.52 | 17.59 | 25.29 | 17.31 | 21.55 | 26.86 | 15.48 | 20.23 |
| Point-Cache | 20.82 | 18.53 | 26.86 | 22.48 | 25.33 | 28.42 | 19.64 | 23.15 |
| PTA | **21.44** | **18.63** | **29.53** | **25.40** | **28.83** | **30.92** | **20.12** | **24.98** |

