# OpenReview forum: "Prototype-Based Test-Time Adaptation of Vision-Language Models"
_ICML.cc/2026/Conference — ICML 2026 regular_

### Official Review · Reviewer_bBYQ · 2026-03-10

**Soundness:** 3
**Presentation:** 3
**Significance:** 3
**Originality:** 3
**Overall Recommendation:** 4
**Confidence:** 4

**Summary:**

This paper proposes Prototype-Based Test-Time Adaptation (PTA), a method that uses neither backpropagation nor cache memory for the test-time adaptation of vision-language models. The authors completely eliminate the external memory retrieval process, opting instead to accumulate streaming data into class-specific knowledge prototypes. It presents an efficient representation learning framework that maintains a stable joint embedding space across modalities without relying on heavy backpropagation or external LLM prompts.

**Compliance With Llm Reviewing Policy:**

Affirmed.

**Final Justification:**

The authors have provided satisfactory responses to my inquiries, effectively addressing my concerns. Consequently, I will retain my positive score.

**Key Questions For Authors:**

1. Is there a filtering mechanism to prevent the accumulation of incorrect visual features into the prototype when extreme Out-of-Distribution samples are introduced where the initial CLIP predictions point to completely wrong answers?
2. Can the model prevent overfitting of the corresponding class prototypes even in highly imbalanced stream scenarios, such as when data from only 1 or 2 specific classes is fed 1000 times consecutively?

**Limitations:**

yes

**Strengths And Weaknesses:**

Strengths:
1. It prevents the exponential growth of external memory (cache) requirements that typically occurs as the number of classes increases.
2. It stably prevents the representation collapse that occurs when accumulating purely visual knowledge by utilizing text-anchoring.
3. The model demonstrates consistent performance improvements regardless of the quality or detail level of the text prompts, ranging from a single Vanilla prompt to detailed GPT-generated prompts.

Weaknesses:
1. If the 'adaptive forgetting' or reset strategy mentioned in the limitations were to be introduced, what metrics would be used to identify outdated knowledge?
2. Although the adaptive weight calculation method is intuitive, it lacks theoretical justification for why it should be an exponential function compared to other alternatives like linear scaling.
3. Since the contribution score, which forms the basis for updates, relies entirely on the original CLIP's zero-shot confidence, there is a concern that errors could be amplified in target domains where the zero-shot performance is severely degraded.

---

> ### Author Rebuttal · Authors · 2026-03-30
>
> **We sincerely appreciate your careful review and positive feedback. Please kindly see our responses to your comments below.**
>
> **Q1: What metrics would be used to identify outdated knowledge?**
>
> **A1:** Thank you for this insightful question. **A simple way to identify outdated knowledge is to monitor the recent update activity of each class prototype using a sliding window**. If a prototype has been active before but its update count in the current window is below a small threshold, we regard it as outdated and trigger a reset/forgetting operation. We further examine this idea under highly imbalanced streams in **A5**, where the results support recent prototype activity as a useful metric for identifying outdated knowledge.
>
> **Q2: Why choose an exponential adaptive weight over linear scaling?**
>
> **A2:** Thank you for the comment. **We agree that the exponential form is not the only possible choice, and we do not claim strict theoretical optimality for it.** Our motivation is based on the properties needed for stable prototype accumulation in streaming: **(1) monotonicity**, so higher-confidence samples contribute more, **(2) boundedness**, to avoid overly aggressive updates, and **(3) smooth saturation**, so that a few early high-confidence samples do not dominate the prototype update. The exponential mapping naturally satisfies these properties, whereas linear scaling provides weaker suppression of high-confidence samples.
>
> To further validate this design choice, we compare **linear scaling ($\beta = s/h$), sigmoid ($\beta = \sigma((s-\mu)/h)$), and exponential ($\beta = 1 - e^{-s/h}$)** adaptive weights. The results show that the exponential weight performs best, outperforming both linear and sigmoid mappings. This supports it as a better practical choice for stable prototype accumulation. We list the results below.
> Method|ImageNet|10 Cross-Domain
> -|-|-
> Linear|69.9|69.0
> Sigmoid|69.6|68.4
> Exponential|**70.3**|**69.4**
>
> **Q3: Error amplification under severely degraded zero-shot performance.**
>
> **A3:** Thank you for this important comment. We clarify that **PTA does not use baseline pseudo-labels as hard labels for prototype updates, but rather class-wise confidence as a soft weighting signal for smooth accumulation**. Thus, even if the prediction is wrong, the update remains smooth, minimizing the risk of severe error accumulation. Additionally, text anchoring keeps prototype accumulation aligned with a stable semantic space, which further limits error amplification. As a result, PTA does not fail on tasks with very low zero-shot performance. For instance, on EuroSAT, where the zero-shot accuracy is below 50%, PTA improves CLIP by +13.9%.
>
> **Q4: Is there a filtering mechanism to prevent the accumulation of incorrect visual features into the prototype?**
>
> **A4:** Thank you for this important question. We first clarify that PTA does not use an explicit hard filtering mechanism before prototype updates. More importantly, **PTA does not use the top-1 pseudo-label as a hard update target. Instead, it uses the full class-wise confidence distribution as a soft weighting signal for smooth prototype accumulation.** Therefore, even when the initial prediction is wrong under an extreme OOD shift, the update is not assigned to a single class in an extreme manner, which reduces the risk of severe prototype poisoning.
>
> In addition, **prototype accumulation is intended to maintain a compact feature state rather than a sparse sample memory/cache.** Therefore, aggressive filtering is less suitable for prototype-based TTA, since it may make the accumulated state sparse and unstable.
>
> **Q5: Can PTA avoid prototype overfitting under highly imbalanced streams?**
>
> **A5:** Thank you for this question. In response, we evaluate PTA under the Separate setting in [a], where **classes arrive sequentially**. Under this highly imbalanced stream, PTA (w=0.001,h=1000) outperforms both CLIP (67.5 vs. 65.1) and the cache-based TDA (67.5 vs. 66.6), indicating that **PTA is not fragile under highly imbalanced streams and is less prone to prototype overfitting**. In addition, we include the window-based reset variant (window size = 50) from **A1**, which further improves the average accuracy from 67.5 to 70.0. This suggests that **PTA equipped with forgetting/reset has the potential to serve as a strong baseline for TTA under highly imbalanced streams**, while we leave more advanced designs to future work. The results are listed below. ([a] Realistic Test-Time Adaptation of Vision-Language Models)
> Method|ImageNet|Flower|DTD|Pets|Cars|UCF|Caltech|Food|SUN|Aircraft|Eurosat|Avg
> -|-|-|-|-|-|-|-|-|-|-|-|-
> CLIP|66.6|70.7|43.5|89.1|65.6|67.5|93.2|85.9|62.5|24.7|48.3|65.1
> TDA|67.4|72.3|45.0|88.9|65.9|69.6|93.6|85.2|64.6|24.9|55.3|66.6
> PTA|68.5|73.2|45.6|90.6|68.5|71.1|93.7|85.6|66.7|25.5|53.9|67.5
> PTA (Reset)|79.3|74.3|47.0|91.2|72.7|71.9|94.6|88.4|72.7|25.6|52.1|**70.0**
>
> **We will include the added results and discussions in the final version.**

---

> > ### Author Rebuttal · Reviewer_bBYQ · 2026-04-02
> >
> > Thank you to the authors for the detailed rebuttal and the additional experiments. I have carefully read your responses. I have decided to maintain my original positive score.

---

> > > ### Author Response · Authors · 2026-04-02
> > >
> > > Thank you very much for your careful reading and positive feedback. We sincerely appreciate your recognition of the key strengths of our work, including the cache-free scalability, the stabilizing effect of text anchoring, and the robustness across different prompt qualities. We are also grateful for your positive assessment of our rebuttal and additional experiments. Your support means a lot to us.

---

### Official Review · Reviewer_3qkr · 2026-03-12

**Soundness:** 3
**Presentation:** 3
**Significance:** 3
**Originality:** 3
**Overall Recommendation:** 4
**Confidence:** 3

**Summary:**

The paper addresses the challenge of TTA for VLMs. Previous backpropagation-free TTA methods suffer from two major bottlenecks: inference latency scales poorly as the cache grows with the number of classes, and performance degrades when the cache accumulates noisy or insufficient samples. To solve this, the authors propose PTA to dynamically maintain and update a single feature prototype for each class during the testing phase. The proposed approach accumulates knowledge from test examples through class-specific prototypes and reduces the memory footprint and inference latency compared to instance-caching methods. PTA demonstrates highly competitive performance on various benchmarks.

**Compliance With Llm Reviewing Policy:**

Affirmed.

**Final Justification:**

I thank the authors for their detailed response. My questions have been adequately addressed, and I have no further comments. Based on the contributions of the paper and the feedback from other reviewers, I would like to raise my recommendation to “Weak Accept.”

**Key Questions For Authors:**

1. How does PTA perform if the test data arrive in a highly skewed or temporally correlated distribution? For example, if the system sees more instances of one class over another class, will the prototype drift to one class?
2. How sensitive is the method to the confidence threshold used for prototype updating? Was this threshold set universally across all datasets, or did it require per-dataset tuning?
3. If an out-of-distribution shift causes a confidently incorrect prototype update early in the test stream, will it permanently poison that class's prototype?

**Limitations:**

The authors have addressed technical limitations in the Appendix. Test-time adaptation allows models to change their behavior post-deployment. Although this improves accuracy, it introduces the risks of unpredictability and bias amplification. For example, if deployed in a demographic or medical setting, the model might adapt its prototypes based on spurious correlations in the immediate local environment, leading to unequal performance between different user groups. This has not been noted in the paper.

**Strengths And Weaknesses:**

Strengths: (1) The motivation of the paper is very clear. The framework is developed around the twin issues of cache-induced latency and noisy sample corruption, which justifies why a prototype-based method is necessary. (2) The mathematical and structural transition from instance-level caching to class-level prototypes is sound and logically solves the search complexity problem of standard cache retrieval methods. The use of a confidence threshold to gate prototype updates is a well-established technique to prevent confirmation bias in pseudo-labeling. (3) The proposed PTA is very helpful for the deployment of VLMs on edge devices or in real-time streaming applications. (4) Applying prototype as a lightweight, backprop-free replacement for cache-retrieval in VLM test-time adaptation is a clever idea.

Weaknesses: (1) The paper lacks a clear ablation on how the tuning was performed without violating strict test-time assumptions. (2) The reliability of confidence-gating is highly dependent on the VLM being well-calibrated. If the base VLM is overly confident but incorrect on a severely out-of-distribution domain, PTA could still suffer catastrophic error propagation, as bad prototypes will reinforce bad predictions. (4) The practical significance may be reduced if the absolute accuracy gains over simpler zero-shot baselines on certain datasets are marginal, although the efficiency gains remain valid. (5) The novelty of this approach lies more in its application to this specific VLM bottleneck than in its fundamental algorithmic breakthrough.

---

> ### Author Rebuttal · Authors · 2026-03-31
>
> **Q1: How was tuning done without violating strict test-time assumptions?**
>
> **A1:** Thank you for this important comment. We clarify that **PTA does not perform any per-dataset tuning on test data**. In the main experiments, **the hyperparameters are fixed universally across benchmarks (e.g., h=20,w=0.01)**, and the update rule is applied fully online without using future samples or test labels. The ablations are intended only as sensitivity analyses to show robustness, rather than as per-dataset tuning or model selection on the test stream. In this sense, PTA follows the standard strict test-time adaptation protocol without violating test-time assumptions.
>
> **Q2: Can overconfident incorrect predictions cause catastrophic error propagation in PTA, and will it permanently poison the prototypes?**
>
> **A2:** Thank you for this important question. We agree that overconfident wrong predictions may pose a risk. However, **PTA mitigates this by not using the top-1 pseudo-label as a hard update target. Instead, it uses the full class-wise confidence distribution as a soft weighting signal, so even when predictions are wrong, updates are not assigned in an extreme manner.** In addition, the prototypes are text-anchored, which further constrains error amplification.
>
> To directly test this, we conducted an **early poisoning stress test** by placing confidently incorrect samples (with confidence > 0.8) at the beginning of the stream. We used prefixes of size 1/5/10/all and compared them to the normal stream order. The results are shown below.
> Setting|ImageNet|ImageNet-V|Flower|DTD|Pets|Cars|UCF|Caltech|Food|SUN|Aircraft|Eurosat|Avg
> -|-|-|-|-|-|-|-|-|-|-|-|-|-
> 1|70.3|64.9|75.0|47.6|91.2|68.6|73.2|94.7|86.2|69.1|26.0|61.2|69.0
> 5|70.2|64.9|74.9|47.6|91.4|68.5|73.3|94.7|86.4|69.1|26.0|61.8|**69.1**
> 10|70.3|64.8|75.1|47.6|91.5|68.7|73.1|94.8|86.4|69.2|26.0|61.7|**69.1**
> All|70.3|64.9|75.2|47.5|91.0|68.6|73.0|94.7|86.4|69.3|26.0|61.6|69.0
> Normal|70.3|64.9|75.2|47.7|91.1|68.6|73.1|94.8|86.4|69.2|26.1|61.6|**69.1**
>
> This result shows that the performance remains nearly unchanged under different poisoning prefix sizes (69.0 / 69.1 / 69.1 / 69.0), compared with the normal stream (69.1). This indicates that **even when confidently incorrect samples are placed at the beginning of the stream, PTA does not suffer from catastrophic collapse or permanent prototype poisoning.**
>
> **Q3: Practical significance and novelty.**
>
> **A3:** Thank you for your comment. We would like to clarify that **the practical significance of PTA lies not only in improving accuracy but also in achieving a good balance between performance and efficiency**. Our key innovation is the introduction of a prototype-based TTA paradigm that avoids the delays and information loss caused by cache population and retrieval in previous methods. As a result, PTA achieves effective balance between performance and efficiency. For example, PTA outperforms TDA by 1.4% on cross-domain benchmarks, and on ImageNet-1K, PTA retains 92% of the baseline throughput, while the previously recognized efficient baseline TDA retains only 50%.
>
> **Q4: How does PTA perform if the test data arrive in a highly skewed or temporally correlated distribution?**
>
> **A4:** Thank you for this question. In response, we evaluate PTA under the Separate setting in [a], **where classes arrive sequentially, which serves as an extreme stress test for highly skewed and temporally correlated streams.** Under this setting, PTA outperforms both CLIP (67.5 vs. 65.1) and TDA (67.5 vs. 66.6), indicating that **PTA does not suffer from drift-induced collapse even when one class dominates the stream for a period of time**. The results are listed below. ([a] Realistic Test-Time Adaptation of Vision-Language Models)
> Method|ImageNet|Flower|DTD|Pets|Cars|UCF|Caltech|Food|SUN|Aircraft|Eurosat|Avg
> -|-|-|-|-|-|-|-|-|-|-|-|-
> CLIP|66.6|70.7|43.5|89.1|65.6|67.5|93.2|85.9|62.5|24.7|48.3|65.1
> TDA|67.4|72.3|45.0|88.9|65.9|69.6|93.6|85.2|64.6|24.9|55.3|66.6
> PTA|68.5|73.2|45.6|90.6|68.5|71.1|93.7|85.6|66.7|25.5|53.9|**67.5**
>
> **Q5: Confidence threshold sensitivity and setting.**
>
> **A5:** Thank you for the question. We would like to clarify that **PTA does not rely on a hard confidence threshold for prototype updating**. Instead, it uses the full class-wise confidence distribution as a soft weighting signal for smooth prototype accumulation, which allows all samples to contribute. We also emphasize that the hyperparameters of PTA are kept fixed across all datasets.
>
> **Q6: Limitation.**
>
> **A6:** Thank you for raising this important issue. We agree that TTA may introduce unpredictability and bias amplification after deployment, which remains an open issue in the TTA community. While our paper focuses on an efficient backpropagation-free TTA method, we acknowledge the need to discuss its ethical and fairness implications in real-world use. We will explicitly add this point to the Limitations section in the revision.

---

> > ### Author Rebuttal · Reviewer_3qkr · 2026-04-03
> >
> > I thank the authors for their detailed response. My questions have been adequately addressed, and I have no further comments. Based on the contributions of the paper and the feedback from other reviewers, I would like to raise my recommendation to “Weak Accept.”

---

> > > ### Author Response · Authors · 2026-04-03
> > >
> > > Thank you very much for your thoughtful reconsideration and for raising your recommendation to Weak Accept. We are very pleased that our responses adequately addressed your questions. We also sincerely appreciate your recognition of the clear motivation of our work, the sound design of PTA, and its practical value for lightweight real-world deployment. Your support means a lot to us.

---

### Official Review · Reviewer_RmEX · 2026-03-13

**Soundness:** 3
**Presentation:** 3
**Significance:** 2
**Originality:** 2
**Overall Recommendation:** 4
**Confidence:** 3

**Summary:**

This paper proposes Prototype-Based Test-Time Adaptation (PTA), a backpropagation-free and cache-free method for Vision-Language Models (VLMs) that improves efficiency and accuracy during test-time adaptation. PTA replaces external memory with dynamically evolving class-specific prototypes, enabling the model to adapt to incoming data streams without high latency or error accumulation. Experiments on multiple 2D and 3D benchmarks show that PTA achieves state-of-the-art adaptive accuracy while maintaining high inference efficiency.

**Compliance With Llm Reviewing Policy:**

Affirmed.

**Final Justification:**

I maintain my positive score.

**Key Questions For Authors:**

See weakness.

**Limitations:**

See weakness.

**Strengths And Weaknesses:**

Strength：
* The paper is generally well written and easy to follow.
* This paper conducts extensive experiments. The paper not only conducts thorough evaluations on traditional 2D cross-domain and OOD datasets but also impressively extends the framework to the 3D point cloud modality.

Weakness:
* PTA depends on accumulated prototypes. At the beginning of the test stream, the accumulated prototype may be unstable or harmful. In realistic streaming scenarios, accumulated prototype may drift due to the non-iid test stream. For example, under the setting described in \[a], BCP is likely to fail.
* The paper shows inconsistent baseline selection across tables. The strongest backpropagation-free competitor in Table 1 is BoostAdapter (achieving 68.68%, highly competitive with PTA's 69.38%). However, BoostAdapter mysteriously disappears from Table 2 (OOD Generalization) and Table 4 (Efficiency Analysis). Similarly, the strong backpropagation-based method DPE (which actually slightly outperforms PTA in Table 1 with 69.40%) is entirely omitted from the efficiency comparison in Table 4. Omitting the strongest competitors in subsequent critical evaluations is a major red flag.
* I wonder about the performance on the corruption datasets (e.g., ImageNet-C).

\[a] Realistic test-time adaptation of vision-language models.

---

> ### Author Rebuttal · Authors · 2026-03-29
>
> **Q1. Is PTA unstable or harmful at the beginning of the test stream, and can it drift under non-iid streaming scenarios?**
>
> **A1.** Thank you for this valuable comment. In response, we add two new experiments to evaluate PTA from the perspectives of **performance at the beginning of the stream** and **robustness under non-iid drift.**
> - **Exp1:** We evaluate PTA using only the first 10/20/50/100 samples, and report the average accuracy over 11 datasets, including ImageNet and 10 datasets in cross-domain benchmark.
> - **Exp2:** Following the setting of [a], we evaluate the robustness of PTA under non-iid streaming scenarios. For this more challenging setting, we use a more conservative configuration (h=1000,w=0.001). ([a] Realistic test-time adaptation of vision-language models)
>
> In **Exp1**, PTA does not hurt performance at the beginning of the stream: with 10 samples, it matches the baselines. With 20 samples, PTA begins to show gains, and the benefit becomes clearer at 50 and 100 samples, where it outperforms both CLIP and TDA on average. **This shows that prototype accumulation is not harmful early on and becomes useful after only a few observations.**
>
> In **Exp2**, we observe that as the stream becomes increasingly non-iid, PTA remains stable (67.8, 67.4, 67.5, and 67.5), consistently outperforming both CLIP and TDA on average, with a larger advantage under stronger non-iidness. **This indicates that PTA remains robust under non-iid drift and degrades more gracefully than the cache-based TDA.**
>
> In summary, these results show that **PTA is neither harmful at the beginning of the stream nor fragile under the non-iid streaming scenarios.** We list the results below.
> - **Tab 1: Average accuracy using only the first 10/20/50/100 samples**
> Method|10|20|50|100
> -|-|-|-|-
> CLIP|**81.0**|74.0|68.3|67.0
> TDA|**81.0**|74.0|68.3|67.2
> PTA|**81.0**|**74.5**|**69.9**|**68.7**
> - **Tab 2: Results under non-iid streaming scenarios with different $\gamma$, where smaller $\gamma$ indicates stronger non-iidness. In the Separate setting, classes arrive sequentially**
> $\gamma$|Method|ImageNet|Flower|DTD|Pets|Cars|UCF|Caltech|Food|SUN|Aircraft|Eurosat|Avg
> -|-|-|-|-|-|-|-|-|-|-|-|-|-
> -|CLIP|66.6|70.7|43.5|89.1|65.6|67.5|93.2|85.9|62.5|24.7|48.3|65.1
> 0.1|TDA|68.3|72.5|45.5|89.6|66.9|71.0|93.4|86.1|66.0|25.4|60.6|67.7
> 0.1|PTA|67.2|74.2|46.3|90.5|67.2|71.1|93.7|86.3|66.9|25.6|57.1|**67.8**
> 0.01|TDA|68.2|72.6|45.2|89.3|66.5|70.1|93.5|85.8|66.2|25.2|56.5|67.1
> 0.01|PTA|67.5|73.3|45.8|90.6|67.7|70.9|93.8|86.1|65.8|25.5|54.8|**67.4**
> 0.001|TDA|67.9|72.5|45.1|89.0|66.3|69.7|93.6|85.5|65.1|25.1|55.3|66.8
> 0.001|PTA|68.1|73.3|45.6|90.6|68.2|70.9|93.8|85.8|66.6|25.5|54.1|**67.5**
> Separate|TDA|67.4|72.3|45.0|88.9|65.9|69.6|93.6|85.2|64.6|24.9|55.3|66.6
> Separate|PTA|68.5|73.2|45.6|90.6|68.5|71.1|93.7|85.6|66.7|25.5|53.9|**67.5**
>
> **Q2: BoostAdapter and DPE missing from some tables.**
>
> **A2:** Thank you for pointing this out. We agree that the baseline selection across tables should be consistent. We therefore reproduced the missing results of BoostAdapter and DPE. We find that BoostAdapter achieves higher OOD accuracy than PTA. It is a strong baseline that expands the cache with additional augmented views. However, this comes at the cost of lower efficiency. In contrast, **the key innovation of PTA is a cache-free paradigm for efficient TTA, which accumulates knowledge directly in prototypes and thereby eliminates the overhead of cache population and retrieval.** For example, PTA reaches 81.8 FPS, whereas BoostAdapter achieves only 37.5 FPS. In addition, **PTA remains effective without augmented views in the cross-domain and point-cloud settings.** The added results are listed below.
> - **Tab 3: Efficiency comparison on ImageNet**
> Method|FPS (Image/Second)|Memory (Mib)
> -|-|-
> DPE|3.8|6538
> BoostAdapter|37.5|834
> PTA|**81.8**|**744**
> - **Tab 4: Comparison on OOD generalization**
> Method|I|I-A|I-V|I-R|I-S|Avg
> -|-|-|-|-|-|-
> BoostAdapter|69.4|62.9|65.0|80.7|51.3|65.9
> PTA|70.3|61.2|64.9|80.8|51.0|65.6
>
> **Q3: PTA’s performance on corruption datasets such as ImageNet-C.**
>
> **A3.** Thank you for the suggestion. We add PTA’s results on **ImageNet-C at corruption severity level 5**, which is the most challenging setting. PTA achieves the best average accuracy (26.3%), outperforming CLIP (25.7%) and TDA (25.8%). We also note that the paper already reports strong results on **point-cloud corruption benchmarks**. The added results are listed below.
> - **Tab 5: Results on ImageNet-C**
> Method|Gaussian|Shot|Impulse|Defocus|Glass|Motion|Zoom|Snow|Frost|Fog|Brightness|Constrast|Elastic|Pixelate|JPEG|Avg
> -|-|-|-|-|-|-|-|-|-|-|-|-|-|-|-|-
> CLIP|13.0|14.4|13.7|24.1|15.8|25.0|26.6|32.6|30.7|37.0|55.0|17.6|13.6|33.0|33.5|25.7
> TDA|9.6|11.5|11.2|24.8|14.6|23.9|25.0|35.1|32.8|38.7|56.5|16.3|14.2|38.8|33.8|25.8
> PTA|12.9|15.2|13.9|24.5|15.8|25.6|26.3|33.6|32.1|37.4|56.6|17.3|14.4|33.5|35.2|**26.3**
>
> **We will include the added results and discussions in the final version.**

---

> > ### Author Rebuttal · Reviewer_RmEX · 2026-04-03
> >
> > I would like to thank the authors for the rebuttal. I maintain my positive score.

---

> > > ### Author Response · Authors · 2026-04-03
> > >
> > > Thank you very much for your positive feedback and for maintaining your supportive score. We sincerely appreciate your recognition that the paper is clearly written and easy to follow, as well as your acknowledgment of our extensive evaluations across both 2D settings and the 3D point cloud modality. Your support means a lot to us.

---

### Official Review · Reviewer_UZYY · 2026-03-13

**Soundness:** 2
**Presentation:** 3
**Significance:** 1
**Originality:** 1
**Overall Recommendation:** 2
**Confidence:** 5

**Summary:**

This paper proposes a Prototype-Based Test-Time Adaptation (PTA) for vision–language models (VLMs) under distribution shift. Instead of storing previously observed test samples in a memory cache, the method maintains a set of class-specific prototypes that are updated online during inference. For each incoming test sample, visual features extracted from the model are combined with predicted class probabilities to update the corresponding class prototype through an exponential moving average mechanism. The adapted prototypes are then integrated with the original zero-shot classifier of the VLM to refine predictions. The study strives to investigate the context of efficient test-time adaptation by replacing sample-level memory with a more compact prototype representation.

To evaluate the proposed method, experiments are conducted on multiple benchmarks for both image classification and point cloud recognition. The results show that the proposed approach achieves competitive or improved performance compared with several recent test-time adaptation methods, particularly those that avoid gradient-based optimization during inference.

**Compliance With Llm Reviewing Policy:**

Affirmed.

**Final Justification:**

The author partially addressed my questions, but I still do not agree with their arguments regarding the problem formulation and methodological innovation. Therefore, I maintain my original score.

**Key Questions For Authors:**

See Strengths And Weaknesses.

**Limitations:**

Yes, the authors put the "limitation" part into their appendix.

**Strengths And Weaknesses:**

Pros:
1. The paper is generally well organized and clearly written.

2. The proposed PTA, which is updated based on exponential moving averages and confidence weighting, is straightforward and computationally efficient. The method avoids gradient-based optimization at test time, which makes it easy to implement and potentially attractive for deployment scenarios where backpropagation-based TTA is impractical.

3. Experiments are conducted on multiple distribution shift benchmarks and include comparisons with several recent backpropagation-free TTA baselines. The reported results are generally consistent across datasets and suggest that the method can provide modest improvements in certain settings.

Cons:
1. The method explicitly updates prototypes using incoming test samples during inference, effectively accumulating information from the test distribution. This blurs the distinction between test-time adaptation and unsupervised domain adaptation. If the method already relies on test samples (not a single test, not the same to TPT) for updating internal representations, it is unclear why the problem should still be framed strictly as test-time adaptation rather than a form of adaptation using test data. While several recent works in the literature adopt similar settings, the existence of such practices does not necessarily justify the formulation itself. In other words, the fact that prior methods also leverage test samples during inference does not automatically make this problem setting well-motivated. The paper would benefit from a clearer conceptual discussion explaining why this formulation should be considered a principled test-time adaptation scenario rather than a variant of test-distribution-driven adaptation.

2. Multiple prior work already uses memory mechanisms based on test samples (e.g., caches, feature banks, or prototype aggregation). The proposed approach mainly replaces a sample-level memory with prototype representations updated via EMA. The underlying idea of leveraging previously observed test samples remains largely the same, making the conceptual contribution relatively incremental.

3. The method is primarily heuristic and the paper does not provide deeper analysis explaining when or why prototype-based aggregation should be preferable to existing memory-based approaches. In addition, the empirical improvements over recent baselines are generally modest, which further limits the overall significance of the contribution.

4. The prototype update rule relies on the model’s predicted probabilities to accumulate information from test samples. However, if early predictions are incorrect, these errors may be propagated into the prototypes and subsequently influence future predictions. The paper does not analyze how robust the method is to such error accumulation, nor does it provide ablation studies examining this potential failure mode.

5. The evaluation assumes a streaming test scenario where the model can continuously update prototypes using previously observed test samples. However, the paper does not sufficiently discuss the implications of this assumption. For example, it remains unclear how sensitive the method is to the ordering of test samples or whether the improvements persist in settings where cross-sample adaptation is restricted. A more thorough discussion of these assumptions would strengthen the empirical evaluation.

---

> ### Author Rebuttal · Authors · 2026-03-31
>
> **Q1: Why is PTA considered a principled TTA scenario rather than a variant of test-distribution-driven adaptation.**
>
> **A1:** Thank you for the question. We believe **the key distinction between TTA and unsupervised domain adaptation (UDA) lies in whether test samples are used online and causally, or offline as part of a dataset**. PTA follows the standard online TTA protocol used in recent literature [1]: **(1)** no access to source data, **(2)** no target labels, **(3)** no offline training on the target dataset, **(4)** no repeated passes over the test data, and **(5)** no access to future samples when updating the current prediction. Under this protocol, **the model never sees the whole test set in advance**, and adaptation is performed in a **streaming, one-pass, post-deployment manner**. This is different from typical UDA or broader test-distribution-driven adaptation, where the target data are usually available offline as a dataset and can be used for multi-step optimization. We will revise the paper to make this distinction clearer.
>
> **Q2: Is PTA merely an incremental heuristic replacement of sample memory, and why should prototype-based aggregation be preferable to existing memory-based approaches?**
>
> **A2:** Thank you for the comment. We acknowledge that the general idea of leveraging previously observed test samples is not new. However, existing methods mainly operate at the **instance level** (e.g., caches or feature banks), which introduces two practical issues: (1) inference latency grows with cache size, and (2) performance can degrade significantly when the cache is sparse or contains incorrect samples. In contrast, **the key innovation of PTA is to perform adaptation at the prototype level**, which avoids the overhead of cache population and retrieval while maintaining effective test-time adaptation. Thus, PTA achieves an effective balance between performance and efficiency. For example, PTA outperforms the cache-based TDA by 1.4% on the cross-domain benchmark, and on ImageNet, PTA retains 92% of the baseline throughput, whereas the previously recognized efficient baseline TDA retains only 50%.
>
> **Q3: What is the impact of early-stage prediction errors on long-term prototype stability and overall performance?**
>
> **A3:** Thank you for this important comment. We agree that early incorrect predictions could, in principle, lead to error accumulation. However, PTA mitigates this risk in two ways. First, **PTA does not use the top-1 pseudo-label as a hard update target, instead, it uses the full class-wise confidence distribution as a soft weighting signal, so even incorrect predictions do not cause extreme updates to a single class**. Second, the prototypes are text-anchored, which further constrains error amplification.
>
> To directly examine this failure mode, we add an **early poisoning stress test** by moving confidently incorrect samples (confidence > 0.8) to the beginning of the stream, as these samples may introduce strong initial bias that could propagate through prototype updates. We use poisoned prefixes of size 5/10/all and compare them with the normal stream order. The results are shown below.
> Setting|ImageNet|ImageNet-V|Flower|DTD|Pets|Cars|UCF|Caltech|Food|SUN|Aircraft|Eurosat|Avg
> -|-|-|-|-|-|-|-|-|-|-|-|-|-
> 5|70.2|64.9|74.9|47.6|91.4|68.5|73.3|94.7|86.4|69.1|26.0|61.8|69.1
> 10|70.3|64.8|75.1|47.6|91.5|68.7|73.1|94.8|86.4|69.2|26.0|61.7|69.1
> All|70.3|64.9|75.2|47.5|91.0|68.6|73.0|94.7|86.4|69.3|26.0|61.6|69.0
> Normal|70.3|64.9|75.2|47.7|91.1|68.6|73.1|94.8|86.4|69.2|26.1|61.6|69.1
>
> This result shows that the performance remains nearly unchanged under different poisoning prefix sizes, compared with the normal stream. This indicates that **even when confidently incorrect samples are placed at the beginning of the stream, PTA does not suffer from catastrophic collapse or permanent prototype poisoning.**
>
> **Q4: How sensitive is the method to the ordering of test samples in the streaming scenario, and does it remain effective if cross-sample adaptation is restricted?**
>
> **A4:** Thank you for your comment. The definition of the streaming test scenario is discussed in **A1**. Regarding sample ordering, all results are averaged over three random shuffles, and **Table 8 (Appendix)** reports a standard deviation below ±0.15%, indicating **low sensitivity to ordering**. We also examine PTA in more extreme non-iid settings, following [2], to test its robustness under highly imbalanced cross-sample adaptation. The results show PTA works well in challenging scenarios, as shown below  (See **Reviewer RmEX, A1**, for details).
> Method|ImageNet|Flower|DTD|Pets|Cars|UCF|Caltech|Food|SUN|Aircraft|Eurosat|Avg
> -|-|-|-|-|-|-|-|-|-|-|-|-
> CLIP|66.6|70.7|43.5|89.1|65.6|67.5|93.2|85.9|62.5|24.7|48.3|65.1
> PTA|68.5|73.2|45.6|90.6|68.5|71.1|93.7|85.6|66.7|25.5|53.9|**67.5**
>
> **Reference**
> - [1] Efficient test-time adaptation of vision-language models
> - [2] Realistic test-time adaptation of vision-language models

---

> > ### Author Rebuttal · Reviewer_UZYY · 2026-04-04
> >
> > Thank you for the authors’ response. I appreciate the additional experiments and the effort to address my concerns. However, after carefully reading the replies, especially for the problem formulation, I am not yet convinced by the authors’ arguments.
> >
> > My core concern is not about whether the method follows a specific TTA protocol. The fundamental issue is that using previously observed test samples to refine predictions on subsequent samples in a streaming fashion is not new. This sequential learning paradigm has been extensively explored for decades, from classic visual tracking (where each incoming frame is used to continuously update the tracker) to efficient streaming settings in UDA. Given this rich history, I see no principled reason to reframe this well-established sequential learning idea as a new "online test-time adaptation" problem for vision-language models. The authors' response reiterates the standard TTA protocol but does not provide a compelling justification for why this setting should be treated as distinct from existing streaming adaptation paradigms. In addition, being accepted by recent top conferences does not mean a particular problem formulation is automatically correct or well-motivated.
> >
> > Second, the authors acknowledge that leveraging previously observed test samples is not new. The proposed prototype-based aggregation is essentially a heuristic replacement of one memory mechanism (cache) with another (prototypes). The conceptual contribution remains incremental, with no theoretical contribution. In fact, this paper is very similar to earlier work in visual tracking. For example, each incoming frame is used as a new template or to update a buffer of templates, and the updated template pool is then used to track the subsequent frame. What the authors have done is apply this well-established sequential learning idea to a new base model (CLIP) with implementation-level changes, without offering new insight into the problem itself. Therefore, the novelty is insufficient for a top-tier venue.
> >
> > Therefore, I keep my original score.

---

> > > ### Author Response · Authors · 2026-04-05
> > >
> > > Thank you for the detailed follow-up. We acknowledge that PTA is sequential in the sense that samples arrive in a stream. However, **we believe there is an important distinction from visual tracking.** As stated in **[1]**, the goal of object tracking is: “Given the initialized state (e.g., position and size) of a target object in a frame of a video, the goal of tracking is to estimate the states of the target in the subsequent frames.” This implies that, in visual tracking, consecutive inputs are tightly correlated in time and context, and usually share the same object or scene background. **As a result, tracking fundamentally depends on the order of data points, and would break down if such temporal/contextual order were arbitrarily shuffled. More broadly, prior sequential learning paradigms are effective precisely because they exploit temporal/contextual relations across adjacent frames or samples.**
> > >
> > > **In our setting, by contrast, adjacent test samples can be semantically unrelated (e.g., one image may be a dog and the next an airplane), with no inherent event-level or temporal coherence between them, and they may even be completely independent.** Therefore, PTA does not exhibit the same strict dependence on sample order as classical sequential learning paradigms such as visual tracking.
> > >
> > > **This distinction is also supported empirically. If PTA truly relied on sequential structure in the same sense as tracking, random reordering would substantially change its behavior.** However, **Table 8 in the Appendix** shows extremely small variance across different random orders (below ±0.15%), and our early poisoning stress test likewise shows negligible performance change under strongly perturbed prefixes. **These results indicate that PTA is streaming but not order-dependent: it accumulates distributional information from previously observed unlabeled test samples, rather than relying on temporal continuity or event context.** In this sense, **PTA studies causal cross-sample adaptation under deployment-time TTA constraints, rather than temporally/contextual dependent sequential learning such as visual tracking.**
> > >
> > > Moreover, **we would like to clarify that PTA is not merely a heuristic reformulation.** Its motivation is tied to specific weaknesses of existing cache-based VLM adaptation methods: (1) cache population and retrieval introduce additional latency and scaling overhead, and (2) heuristic filtering may lead to sparse memory in the early stage, or cause early confidently incorrect samples to remain in memory. **PTA addresses these issues by replacing sparse instance-level cache memory with a compact prototype-level accumulation mechanism.** For example, PTA outperforms the cache-based TDA by 1.4% on the cross-domain benchmark, and on ImageNet, PTA retains 92% of the baseline throughput, whereas the previously recognized efficient baseline TDA retains only 50%. In addition, we further include TDA in the same poisoning stress test (shown below) and observe that PTA remains almost unchanged, while TDA shows a larger drop, indicating that the cache-based design is more sensitive to such early incorrect samples.
> > >
> > >  **In this sense, our contribution is not that sequential adaptation itself is new, but that, under deployment-time constraints, a cache-free, prototype-based design provides a practical and conceptually distinct alternative to existing cache-based VLM adaptation methods, without relying on strict sample order and while remaining robust under noisy, extreme, and arbitrarily stream/sequential conditions.**
> > >
> > > We sincerely appreciate the time and effort you invested in reviewing our paper. We hope this clarification makes the scope and contribution of our work clearer.
> > >
> > > - **Tab: Robustness under early poisoning. All confidently incorrect samples (confidence > 0.8) are placed at the stream prefix.**
> > > Method|ImageNet|ImageNet-V|Flower|DTD|Pets|Cars|UCF|Caltech|Food|SUN|Aircraft|Eurosat|Avg
> > > -|-|-|-|-|-|-|-|-|-|-|-|-|-
> > > PTA|70.3|64.9|75.2|47.7|91.1|68.6|73.1|94.8|86.4|69.2|26.1|61.6|69.1
> > > PTA(all)|70.3|64.9|75.2|47.5|91.0|68.6|73.0|94.7|86.4|69.3|26.0|61.6|**69.0 (-0.1)**
> > > TDA|69.5|64.7|71.5|45.1|89.4|67.0|70.8|94.1|86.3|67.8|25.3|62.5|67.8
> > > TDA(all)|69.0|64.5|70.7|45.2|88.8|66.8|70.9|93.8|85.8|67.1|24.2|61.1|**67.3 (-0.5)**
> > >
> > > **Reference**
> > > - **[1] Online Object Tracking: A Benchmark (CVPR 2013)**

---

### Decision · Program_Chairs · 2026-04-30

**Decision:**

Accept (regular)

**Comment:**

This paper proposes Prototype-Based Test-Time Adaptation, an efficient method for adapting VLMs to test distributions without backpropagation. By maintaining compact class-specific prototypes updated with confidence-weighted EMA, PTA achieves strong performance while preserving high inference efficiency.

The reviewers raised several main concerns, including the level of conceptual novelty, robustness under challenging streaming conditions, completeness of evaluation, and potential risks such as error accumulation and prototype drift. In the rebuttal phase, the authors provided additional experiments and clarifications, including early poisoning tests, non-i.i.d. stream evaluations, and comparisons with missing baselines such as BoostAdapter and DPE. After rebuttal, all reviewers with a score of 4 acknowledged that their concerns had been fully addressed and maintained or strengthened their positive evaluations.

However, reviewer UZYY remained unconvinced, primarily questioning whether the work constitutes a genuinely new formulation or an incremental extension of existing sequential adaptation methods, and kept the reject score. Notably, this reviewer did not provide a final justification after the rebuttal.

Overall, the decision is based on the majority consensus. Three reviewers provided positive evaluations after rebuttal and recognized the method’s technical soundness, empirical strength, and practical efficiency. Considering the lack of a final justification from the dissenting reviewer and the clear positive vote from the majority, the paper is recommended for acceptance, but the authors should revise the manuscript to address the remaining issue, especially the clarification of attribute interpretation.